# Effector membrane translocation biosensors reveal G protein and βarrestin coupling profiles of 100 therapeutically relevant GPCRs

Charlotte Avet[1†], Arturo Mancini[2†], Billy Breton[2‡§], Christian Le Gouill[1‡], Alexander S Hauser[3‡], Claire Normand[2], Hiroyuki Kobayashi[1], Florence Gross[2], Mireille Hogue[1], Viktoriya Lukasheva[1], Stéphane St-Onge[1], Marilyn Carrier[1], Madeleine Héroux[1], Sandra Morissette[2], Eric B Fauman[4], Jean-Philippe Fortin[5], Stephan Schann[6], Xavier Leroy[6*], David E Gloriam[3*], Michel Bouvier[1*]

[1]Institute for Research in Immunology and Cancer (IRIC), and Department of Biochemistry and Molecular Medicine, Université de Montréal, Montréal, Canada; [2]Domain Therapeutics North America, Montréal, Canada; [3]Department of Drug Design and Pharmacology, University of Copenhagen, Copenhagen, Denmark; [4]Internal Medicine Research Unit, Pfizer Worldwide Research, Development and Medical, Cambridge, United States; [5]Pfizer Global R&D, Cambridge, United States; [6]Domain Therapeutics, Illkirch-Strasbourg, France

**\*For correspondence:**
xleroy@domaintherapeutics.com (XL);
david.gloriam@sund.ku.dk (DEG);
michel.bouvier@umontreal.ca (MB)

†These authors contributed equally to this work
‡These authors also contributed equally to this work

**Present address:** §Institute for Research in Immunology and Cancer (IRIC), Université de Montréal, Montréal, Canada

**Abstract** The recognition that individual GPCRs can activate multiple signaling pathways has raised the possibility of developing drugs selectively targeting therapeutically relevant ones. This requires tools to determine which G proteins and βarrestins are activated by a given receptor. Here, we present a set of BRET sensors monitoring the activation of the 12 G protein subtypes based on the translocation of their effectors to the plasma membrane (EMTA). Unlike most of the existing detection systems, EMTA does not require modification of receptors or G proteins (except for $G_s$). EMTA was found to be suitable for the detection of constitutive activity, inverse agonism, biased signaling and polypharmacology. Profiling of 100 therapeutically relevant human GPCRs resulted in 1500 pathway-specific concentration-response curves and revealed a great diversity of coupling profiles ranging from exquisite selectivity to broad promiscuity. Overall, this work describes unique resources for studying the complexities underlying GPCR signaling and pharmacology.

## Editor's evaluation

The authors describe a novel set of biosensors to assess the coupling specificity of 100 therapeutically relevant G proteins-coupled receptors (GPCRs) to various G proteins. The utility of the assay system is well-supported by the data. These tools are likely to be useful for many specific studies of individual receptors, including efforts to discover ligands that display functional selectivity (bias) between G protein pathways or between G proteins and arrestins. The work provides a rich repository of data informing on the possible effector coupling of 100 GPCRs and a set of analytical tools that could guide the development of new drugs, including efforts to discover ligands that display functional selectivity (bias) between G protein pathways or between G proteins and arrestins.

## Introduction

G protein-coupled receptors (GPCRs) play crucial roles in the regulation of a wide variety of physiological processes and represent one-third of clinically prescribed drugs (*Hauser et al., 2017*). Classically, GPCR-mediated signal transduction was believed to rely on linear signaling pathways whereby a given GPCR selectively activates a single G protein family, defined by the nature of its Gα subunit (*Oldham and Hamm, 2008*). Gα proteins are divided into four major families ($G_s$, $G_{i/o}$, $G_{q/11}$, and $G_{12/13}$) encoded by 16 human genes. Once activated, these proteins each trigger different downstream effectors yielding different biological outcomes. It has now become evident that many GPCRs can couple to more than one G protein family and that ligands can selectively promote the activation of different subsets of these pathways (*Namkung et al., 2018*; *Quoyer et al., 2013*). These observations extended the concept of ligand-biased signaling, which was first established for ligand-directed selectivity between βarrestin and G protein (*Azzi et al., 2003*; *Wei et al., 2003*), to functional selectivity between G proteins. Ligand-directed functional selectivity represents a promising avenue for GPCRs drug discovery since it offers the opportunity of activating pathways important for therapeutic efficacy while minimizing activation of pathways responsible for undesirable side effects (*Galandrin et al., 2007*; *Kenakin, 2019*).

To fully explore the potential of functional selectivity, it is essential to have an exhaustive description of the signaling partners that can be activated by a given receptor, providing receptor- and ligand-specific signaling signatures. Currently, few assays allow for an exhaustive pathway-specific analysis of GPCR signaling; these include BRET-based G protein activation sensors platforms (*Galés et al., 2005*; *Masuho et al., 2015*; *Maziarz et al., 2020*; *Mende et al., 2018*; *Olsen et al., 2020*) and the TGF-α shedding assay (*Inoue et al., 2019*). However, several of these platforms require modification of G protein subunits that may create functional distortions. Moreover, these assays may detect non-productive conformational rearrangements of the G protein heterotrimer as was recently reported for $G_{12}$ (*Okashah et al., 2020*).

Here, we describe unique sensors that do not require modification of receptors or G proteins (except for $G_s$) for interrogating the signaling profiles of GPCRs. The platform includes 15 pathway-selective enhanced bystander bioluminescence resonance energy transfer (ebBRET) biosensors monitoring the translocation of downstream effectors to the plasma membrane for $G_{i/o}$, $G_{q/11}$, and $G_{12/13}$, the dissociation of the Gα subunit from the plasma membrane for $G_s$ and the recruitment of βarrestin to the plasma membrane. Overall, the new ebBRET-based **E**ffector **M**embrane **T**ranslocation **A**ssays, named EMTA, provide a readily accessible large scale and comprehensive platform to study constitutive and ligand-directed GPCR signaling. The signaling signatures of 100 GPCRs using the EMTA platform also provides a rich source of information to explore the principles underlying receptor/G protein/βarrestin coupling selectivity relationships. It thus provides a unique set of tools that is complementary to previously described platforms and existing datasets, and offers a map of the coupling potentials for individual GPCR that will stimulate future studies investigating the relevance of these couplings in different physiological systems.

## Results

### ebBRET-based G protein effector membrane translocation assay (EMTA) allows detection of each Gα protein subunit activation

To detect the activation of Gα subtypes, we created an EMTA biosensor platform based on ebBRET (*Namkung et al., 2016*; *Figure 1A*). The biosensors at the heart of EMTA consist of sub-domains of the G protein-effector proteins p63-RhoGEF, Rap1GAP and PDZ-RhoGEF that selectively interact with activated $G_{q/11}$, $G_{i/o}$, or $G_{12/13}$, respectively. These domains were fused at their C-terminus to *Renilla* luciferase (RlucII) and co-expressed with different unmodified receptor and Gα protein subtypes. Upon GPCR activation, the energy donor-fused effectors translocate to the plasma membrane to bind activated Gα proteins, bringing RlucII in close proximity to the energy acceptor, *Renilla* green fluorescent protein, targeted to the plasma membrane through a CAAX motif (rGFP-CAAX), thus leading to an increase in ebBRET. The same plasma membrane translocation principle is used to measure βarrestin recruitment (*Namkung et al., 2016*; *Figure 1B*, top). Because no selective soluble downstream effector of $G_s$ exists, the assay was modified taking advantage of $Gα_s$ dissociation from the plasma membrane following its activation (*Wedegaertner et al., 1996*). In this configuration,

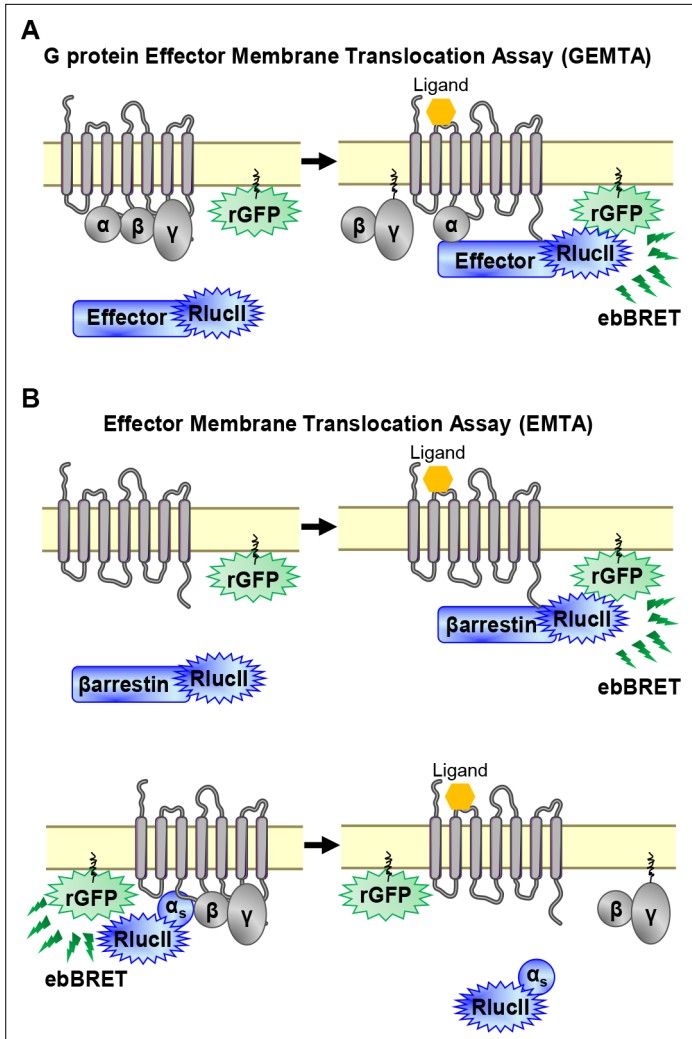

**Figure 1.** EMTA ebBRET platform to monitor G protein activation and βarrestin recruitment. (**A**) Schematic of the G protein Effector Membrane Translocation Assay (GEMTA) to monitor Gα protein activation. Upon receptor activation, RlucII-tagged effector proteins (Effector-RlucII) translocate towards and interact with active Gα subunits from each G protein family, leading to increased ebBRET. (**B**) Principle of the Effector Membrane Translocation Assay (EMTA) monitoring βarrestin recruitment to the plasma membrane (*top*) and Gα$_s$ activation (*bottom*). *Top*; upon receptor activation, RlucII-tagged βarrestins (βarrestin-RlucII) translocate to the plasma membrane, thus increasing ebBRET with rGFP-CAAX. *Bottom*; Internalization of activated RlucII-tagged Gα$_s$ (Gα$_s$-RlucII) following receptor stimulation decreases ebBRET with the membrane-anchored rGFP-CAAX.

the RlucII is directly fused to Gα$_s$ (*Carr et al., 2014*). Its activation upon GPCR stimulation leads to its dissociation from the plasma membrane (*Martin and Lambert, 2016*), resulting in a reduction in ebBRET (*Figure 1B*, bottom).

The sensitivity and selectivity of the newly created G protein EMTA biosensors, were validated using prototypical GPCRs known to activate specific Gα subtypes. The responses were monitored upon heterologous expression of specific Gα subunits belonging to G$_{i/o}$, G$_{q/11}$, or G$_{12/13}$ families in the absence or presence of pharmacological inhibitors and using engineered cells lacking selected Gα subtypes. The dopamine D$_2$ receptor was used to validate the ability of the G$_{i/o}$ binding domain of Rap1GAP (*Jordan et al., 1999*; *Meng et al., 1999*) to selectively detect G$_{i/o}$ activation. The dopamine-promoted increase in ebBRET between Rap1GAP-RlucII and rGFP-CAAX in the presence of Gα$_{i/o}$ subunits was not affected by the G$_{q/11}$-selective inhibitor UBO-QIC (a.k.a., FR900359; *Schrage et al., 2015*; *Figure 2A*, left), whereas the Gα$_{i/o}$ family inhibitor, pertussis toxin (PTX), completely blocked the response for all members of Gα$_{i/o}$ family except for Gα$_z$, known to be insensitive to PTX (*Casey et al., 1990*; *Figure 2A*, right). Gonadotropin-releasing hormone

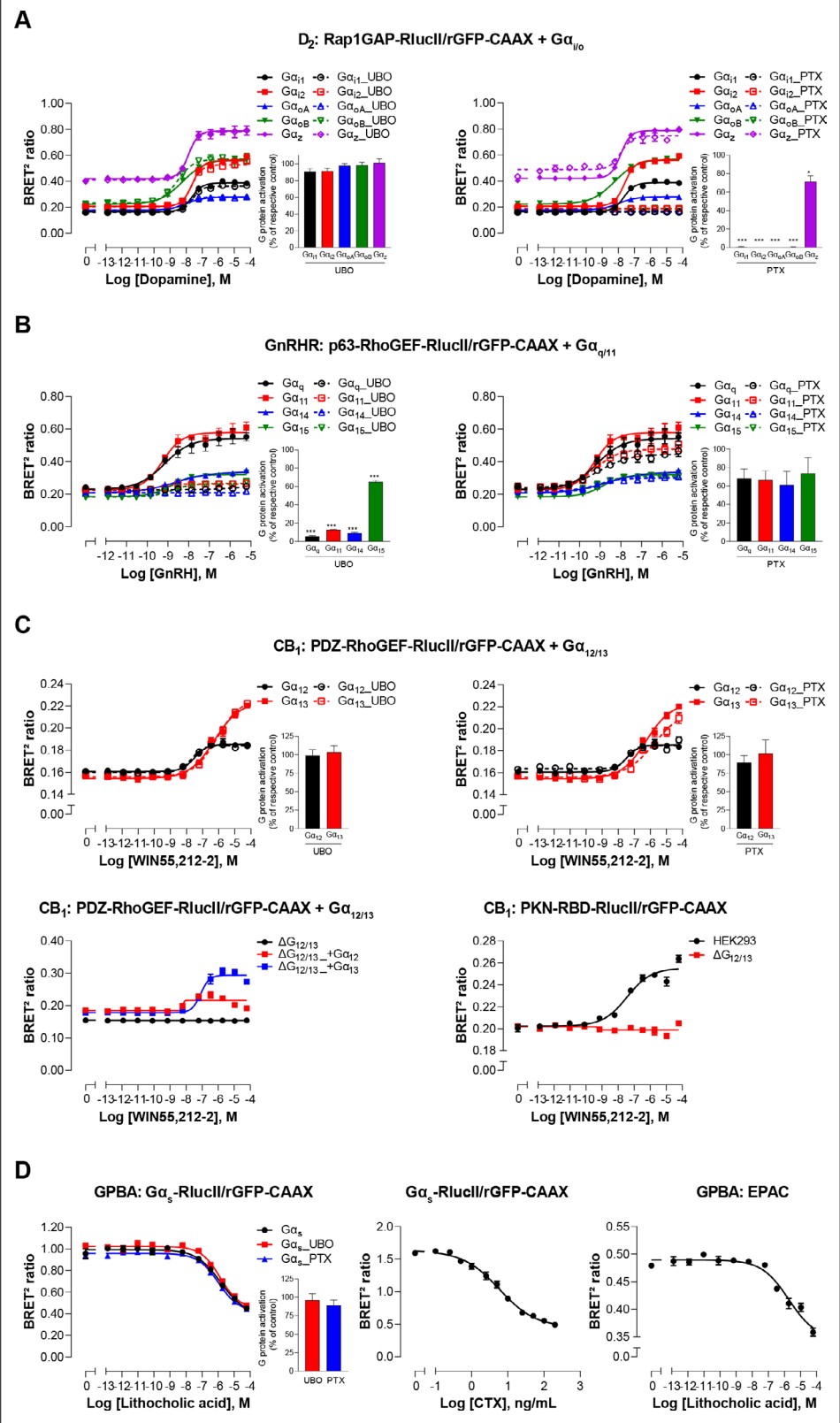

**Figure 2.** Validation of EMTA ebBRET-based platform to monitor Gα protein activation. (**A**) Pharmacological validation of the Gα$_{i/o}$ activation sensor. HEK293 cells were transfected with the D$_2$ receptor and the Gα$_{i/o}$ family-specific sensor, along with each Gα$_{i/o}$ subunit. Concentration-response curve using the Gα$_{i/o}$ activation sensor, in the presence or absence of UBO-QIC (*left*) or PTX (*right*) inhibitors. *Insets*; E$_{max}$ values determined from

*Figure 2 continued on next page*

*Figure 2 continued*

concentration-response curves of inhibitor-pretreated cells. (**B**) Pharmacological validation of the $G\alpha_{q/11}$ activation sensor. HEK293 cells were transfected with the GnRH receptor and the $G\alpha_{q/11}$ family-specific sensor, along with each $G\alpha_{q/11}$ subunit. Concentration-response curve using $G\alpha_{q/11}$ activation sensor, in the presence or absence of UBO-QIC (*left*) or PTX (*right*) inhibitors. *Insets*; $E_{max}$ values determined from dose-response curves of inhibitor-pretreated cells. (**C**) Validation of the $G\alpha_{12/13}$ activation sensor. Cells were transfected with the CB$_1$ receptor and one of the $G\alpha_{12/13}$ activation sensors, along with the $G\alpha_{12}$ or $G\alpha_{13}$ subunits. Concentration-response curves of HEK293 cells (*top*) or the parental and devoid of $G_{12/13}$ ($\Delta G_{12/13}$) HEK293 cells (*bottom*) using the PDZ-RhoGEF-RlucII/ rGFP-CAAX (*top and bottom left*) or PKN-RBD-RlucII/rGFP-CAAX (*bottom right*) sensors, pretreated or not with UBO-QIC or PTX (*top*). (**D**) Pharmacological validation of the $G\alpha_s$ activation sensor. HEK293 cells were transfected with the GPBA receptor and the $G\alpha_s$ activation (*left and central*) or the EPAC (*right*) sensors. *Left*: Concentration-response curves using the $G\alpha_s$ activation sensor in the presence or absence of UBO-QIC or PTX, inhibitors of $G\alpha_q$ or $G\alpha_{i/o}$, respectively. *Central*: Concentration-response activation of the $G\alpha_s$ sensor using CTX, a $G\alpha_s$ activator. *Right*: Concentration-response curve using the EPAC sensor. *Inset*; $E_{max}$ values determined from dose-response curves of inhibitors-pretreated cells. Data are expressed as BRET ratio for the concentration-response curves or expressed in % of respective control cells ($E_{max}$ graphs) and are the mean ± SEM of 3 (**A–C**) or 4 (**D**) independent experiments performed in one replicate. Unpaired t-test (**A–D**): *$p < 0.05$ and ***$p < 0.001$ compared to control cells.

The online version of this article includes the following source data and figure supplement(s) for figure 2:

**Source data 1.** Raw data of *Figure 2*.

**Figure supplement 1.** Influence of endogenous G proteins.

**Figure supplement 1—source data 1.** Raw date of *Figure 2—figure supplement 1*.

**Figure supplement 2.** Validation of EMTA ebBRET-based sensors selectivity for each $G\alpha$ subunit families.

**Figure supplement 2—source data 1.** Raw date of *Figure 2—figure supplement 2*.

**Figure supplement 3.** Influence of G protein, GPCR or effector-RlucII level expression.

**Figure supplement 3—source data 1.** Raw date of *Figure 2—figure supplement 3*.

**Figure supplement 4.** Kinetics of $G\alpha$ proteins and βarrestins recruitment promoted by the ET$_A$ receptor.

**Figure supplement 4—source data 1.** Raw date of *Figure 2—figure supplement 4*.

**Figure supplement 5.** Comparison of EMTA platform and G protein activation BRET assay based on Gαβγ dissociation.

**Figure supplement 5—source data 1.** Raw date of *Figure 2—figure supplement 5*.

**Figure supplement 6.** Western blots of G protein level expression in cells transfected with the EMTA ebBRET platform.

**Figure supplement 6—source data 1.** Original Western blot of *Figure 2—figure supplement 6*.

(GnRH) stimulation of the GnRH receptor (GnRHR), used as a prototypical $G_{q/11}$-coupled receptor, promoted ebBRET between the RlucII-fused $G_{q/11}$ binding domain of p63-RhoGEF (p63-RhoGEF-RlucII; *Lutz et al., 2007*; *Rojas et al., 2007*) and rGFP-CAAX. The ebBRET increase observed in the presence of different $G\alpha_{q/11}$ subunits was not significantly (p = 0.077, 0.0636 and 0.073 for $G_q$, $G_{11}$, and $G_{14}$, respectively) affected by PTX (*Figure 2B*, right), whereas UBO-QIC completely blocked the response for all members of $G\alpha_{q/11}$ family except for $G\alpha_{15}$, known to be insensitive to UBO-QIC (*Schrage et al., 2015*; *Figure 2B*, left). These two G protein-specific EMTA were sensitive enough to detect responses elicited by endogenous G proteins since deletion of $G_{i/o}$ ($\Delta G_{i/o}$) or $G_{q/11}$ ($\Delta G_{q/11}$) subtypes completely abolished the responses induced by $D_2$ or GnRHR activation in the absence of heterologously expressed G proteins (*Figure 2—figure supplement 1I*). It should however be noted that relying on endogenous proteins does not allow the identification of specific members of $G_{i/o}$ (i.e.: $G_{i1}$, $G_{i2}$, $G_{i3}$, $G_{oA}$, $G_{oB}$, or $G_z$) or $G_{q/11}$ (i.e.: $G_q$, $G_{11}$, $G_{14}$, or $G_{15}$) families.

The selectivity of the $G_{12/13}$ binding domain of PDZ-RhoGEF (*Fukuhara et al., 2001*) was confirmed using the cannabinoid receptor type 1 (CB$_1$). The ebBRET between PDZ-RhoGEF-RlucII and rGFP-CAAX in the presence of $G\alpha_{12}$ or $G\alpha_{13}$ promoted by the cannabinoid agonist WIN-55,212–2 was not affected by UBO-QIC (*Figure 2C*, top left), nor PTX (*Figure 2C*, top right). Given the lack of selective $G_{12/13}$ pharmacological inhibitor, we used HEK293 cells genetically deleted for $G\alpha_{12}$ and $G\alpha_{13}$ proteins ($\Delta G_{12/13}$) to further confirm the response selectivity. As expected, PDZ-RhoGEF-RlucII/rGFP-CAAX ebBRET was observed only following reintroduction of either $G\alpha_{12}$ ($\Delta G_{12/13}$_+$G_{12}$) or $G\alpha_{13}$ ($\Delta G_{12/13}$_+$G_{13}$) (*Figure 2C*, bottom left).

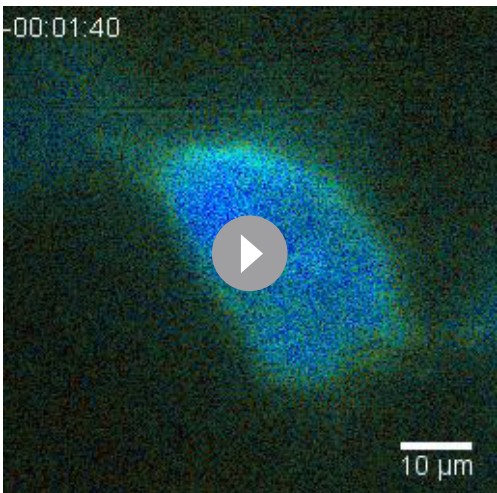

**Video 1.** BRET-based imagery of p63-RhoGEF-RlucII recruitment to the plasma membrane upon $AT_1$ activation. HEK293 cells expressing the p63-RhoGEF-RlucII/rGFP-CAAX sensors with $G\alpha_q$ and $AT_1$ were stimulated with Angiotensin II. BRET levels (the ratio of the acceptor photon count to the total photon count) are expressed as a color code (lowest being black and purple, and highest being red and white).
https://elifesciences.org/articles/74101/figures#video1

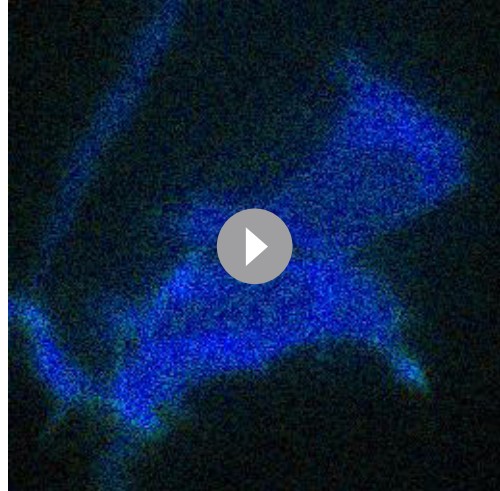

**Video 2.** BRET-based imagery of Rap1GAP-RlucII recruitment to the plasma membrane upon $D_2$ activation. HEK293 cells expressing the Rap1GAP-RlucII/rGFP-CAAX sensors with $G\alpha_{i2}$ and $D_2$ were stimulated with dopamine. BRET levels (the ratio of the acceptor photon count to the total photon count) are expressed as a color code (lowest being black and purple, and highest being red and white).
https://elifesciences.org/articles/74101/figures#video2

The $G_{12/13}$ coupling of $CB_1$ was further confirmed by monitoring the recruitment of PKN to the plasma membrane (*Figure 2C*, bottom right) in agreement with previous reports (*Inoue et al., 2019*).

To further assess the selectivity of each EMTA biosensor, we took advantage of the fact that the endothelin-1 receptor ($ET_A$) can activate $G_{q/11}$, $G_{i/o}$, and $G_{12/13}$ family members. As shown in *Figure 2—figure supplement 2*, only over-expression of the $G\alpha$ family members corresponding to their selective effectors (Rap1GAP for $G_{i/o}$, p63-RhoGEF for $G_{q/11}$, and PDZ-RhoGEF for $G_{12/13}$) significantly increased the recruitment of the effector-RlucII to the plasma membrane. A recent study (*Chandan et al., 2021*) showed that $G_{i/o}$ can also activate full length PDZ-RhoGEF. Although the domain of PDZ-RhoGEF required for this activation has not been identified yet, the selectivity of our PDZ-RhoGEF sensor for $G_{12/13}$ *vs.* all other G protein families most likely results from the fact that we used a truncated version of PDZ-RhoGEF that only contains the $G_{12/13}$ binding domain and lacks the PDZ domain involved in protein-protein interaction, the actin-binding domain and the DH/PH domains involved in GEF activity and RhoA activation (*Aittaleb et al., 2010*).

It should be noted that in the heterologous expression configuration, competition with endogenous G proteins did not occur to a significant extent since the potencies of the responses to a given G protein subtype were not affected by genetic deletion of the different G protein

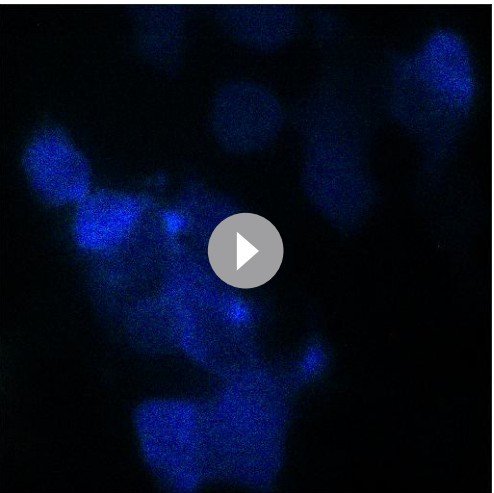

**Video 3.** BRET-based imagery of PDZ-RhoGEF-RlucII recruitment to the plasma membrane upon $TP\alpha R$ activation. HEK293 cells expressing the PDZ-RhoGEF-RlucII/rGFP-CAAX + $G\alpha_{13}$ and $TP\alpha R$ were stimulated with U46619. BRET levels (the ratio of the acceptor photon count to the total photon count) are expressed as a color code (lowest being black and purple, and highest being red and white).
https://elifesciences.org/articles/74101/figures#video3

family members (*Figure 2—figure supplement 1* and *Supplementary file 1A*). Similarly, overexpression of G proteins, GPCRs or effectors-RlucII did not affect the potencies of the responses observed (*Figure 2—figure supplement 3* and *Supplementary file 1B-D*), indicating that, in our experimental conditions, overexpression of the different components of EMTA sensors must likely not bias the coupling response. In addition to spectrometric assessment of coupling selectivity (above) and activation kinetics (*Figure 2—figure supplement 4*), EMTA allows to image the real-time recruitment of the G protein effectors to the plasma membrane (*Videos 1–3*) thus providing spatiotemporal resolution for the imaging detection of $G\alpha_{i/o}$, $G\alpha_{q/11}$, and $G\alpha_{12/13}$ activation.

The sensitivity of the EMTA platform is illustrated by a direct side-by-side comparison of the signals detected with EMTA *vs.* BRET assays based on $G\alpha\beta\gamma$ dissociation ($G\alpha\beta\gamma$) (*Galés et al., 2005*; *Galés et al., 2006*; *Olsen et al., 2020*), that reveals a significantly larger assay windows for EMTA for the 6 $G\alpha$ subunits tested for eight selected receptors, (*Figure 2—figure supplement 5*).

For the $G\alpha_s$ translocation biosensor, the bile acid receptor (GPBA) was chosen for validation (*Kawamata et al., 2003*). As expected, lithocholic acid stimulation resulted in a concentration-dependent decrease in ebBRET between $G\alpha_s$-RlucII and rGFP-CAAX (*Figure 2D*, left). Cholera toxin (CTX), which directly activates $G\alpha_s$ (*De Haan and Hirst, 2004*), led to a decrease in ebBRET (*Figure 2D*, center), confirming that loss of $G\alpha_s$ plasma membrane localization results from its activation. The potency of lithocholic acid to promote $G_s$ dissociation from the plasma membrane was well in line with its potency to increase cAMP production as assessed using a $BRET^2$-based EPAC biosensor (*Leduc et al., 2009*; *Figure 2D*, right). The $G_s$-plasma membrane dissociation ebBRET signal was not affected by UBO-QIC or PTX (*Figure 2D*, left), confirming the selectivity of the biosensor.

## Signaling signatures of one hundred therapeutically relevant receptors reveals distinct G protein and βarrestin selectivity profiles

We used EMTA to assess the signaling signature of a panel of 100 human GPCRs that are either already the target of clinically used drugs (74 receptors), considered for pre- or clinical drug development (6 receptors), or pathophysiologically relevant (*Supplementary file 2A*). To establish the coupling potentials for each receptor, we quantified its ability to activate 15 pathways: $G\alpha_s$, $G\alpha_{i1}$, $G\alpha_{i2}$, $G\alpha_{oA}$, $G\alpha_{oB}$, $G\alpha_z$, $G\alpha_{12}$, $G\alpha_{13}$, $G\alpha_q$, $G\alpha_{11}$, $G\alpha_{14}$, $G\alpha_{15}$ and βarrestin 2 as well as βarrestin 1 and 2 in the presence of GRK2 (*Supplementary file 3*). $E_{max}$ and $pEC_{50}$ values were determined (*Supplementary file 2*) and, based on the pre-determined threshold criteria (Emax ≥mean of vehicle-stimulated +2*SD; see Materials and methods), a 'yes or no' agonist-dependent activation was assigned to each signaling pathway and summarized using radial graph representations (*Figure 3—figure supplement 1*). To assess whether endogenous receptors could contribute to the observed responses, assays were also carried out in cells not transfected with the studied receptor (*Figure 3—figure supplement 2*). When an agonist-promoted response was observed in non-transfected parental HEK293 cells, this response was not considered as a receptor-specific response (see Materials and methods).

To compare the signaling profiles across all receptors and pathways and to overcome differences in receptor expression levels and individual biosensor dynamic windows, we first min-max normalized $E_{max}$ and $pEC_{50}$ values (between 0 and 1) across receptors as a function of a reference receptor yielding the largest response for a given pathway (*Figure 3A*, left). Then, these values were again min-max normalized (between 0 and 1) for the same receptor across pathways, using the pathway with the largest response for this receptor as the reference (*Figure 3A*, right; see description in Materials and methods). Such double normalization allows direct comparison of the coupling efficiency to different G proteins for a given receptor and across receptors for a given G protein. This coupling efficiency is summarized as heatmaps (*Figure 3B*) that reveals a high diversity of signaling profiles. The selectivity toward the different G protein families varies considerably among GPCRs (*Figure 4*). In our dataset, which is the first using unmodified GPCRs and $G\alpha$ proteins (except for $G_s$), 29% of the receptors coupled to only one family, whereas others displayed more promiscuity by coupling to 2, 3, or 4 families (36%, 25%, and 10%, respectively). Receptors coupling to a single G protein family favored the members of the $G_{i/o}$ family. Indeed, 27% of the receptors coupling to $G_{i/o}$ only activated this subtype family in comparison to 0, 2.4 and 9.1% for receptors activating $G_{12/13}$, $G_{q/11}$, and $G_s$, respectively, thus displaying more promiscuous coupling. A detailed comparative analysis of the selectivity profiles that we observed using the EMTA sensors with that of the chimeric G protein-based assay developed by *Inoue et al., 2019* and the IUPHAR/BPS Guide to Pharmacology database (GtP;

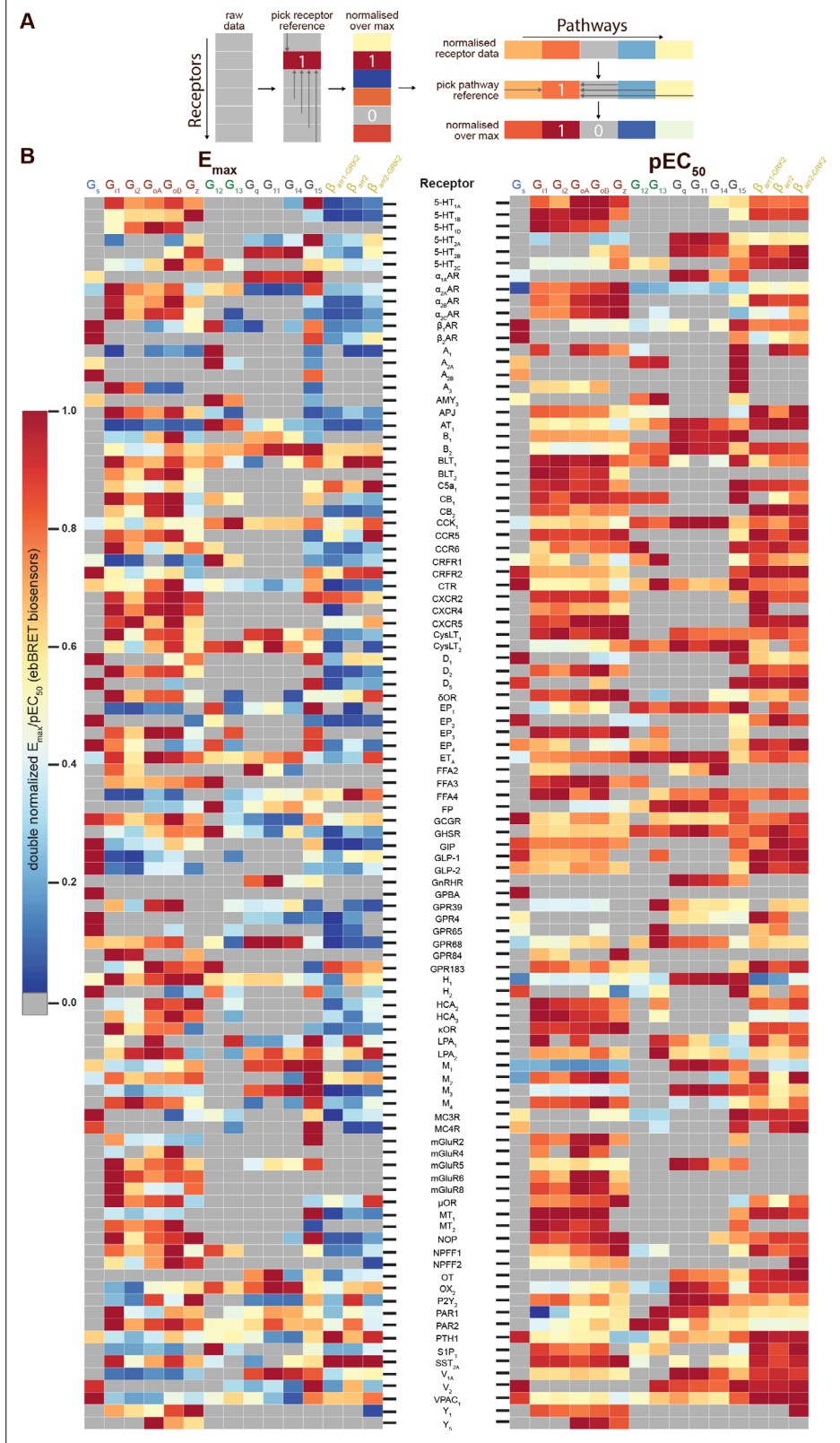

**Figure 3.** Heatmaps illustrating the diversity of receptor-specific signaling signatures detected with the EMTA ebBRET platform. (**A**) First, values within each pathway were normalized relative to the maximal response observed across all receptors (max = 1; *left*). These values were then normalized across pathways for the same receptor, with the highest-ranking pathway serving as the reference (max = 1; *right*). (**B**) Heatmap representation of double

*Figure 3 continued on next page*

*Figure 3 continued*

normalized $E_{max}$ (*left*) and $pEC_{50}$ (*right*) data. Empty cells (grey) indicate no detected coupling. IUPHAR receptor names are displayed.

The online version of this article includes the following source data and figure supplement(s) for figure 3:

**Figure supplement 1.** Receptor-specific signaling signatures.

**Figure supplement 2.** Detection of endogenous receptor-mediated responses with the EMTA ebBRET platform in HEK293 cells.

**Figure supplement 2—source data 1.** Raw data of *Figure 3—figure supplement 2*.

**Figure supplement 3.** Validation of $G_{12/13}$ and $G_{15}$ signaling for the newly characterized GPCRs.

**Figure supplement 3—source data 1.** Raw data of *Figure 3—figure supplement 3*.

---

https://www.guidetopharmacology.org/) is presented in the accompanying paper (*Hauser et al., 2022*). *Supplementary file 2C* allows a direct comparison of the relative potency determined using EMTA for both the new and the already known (i.e.: identified in GtP database) couplings. As can be seen in the table, although in many cases the potency for the novel couplings is lower, this is not a universal finding since for some receptors, the $pEC_{50}$s for the new couplings are similar (ex: $G_{12}$ for $CB_1$; $G_{13}$ for serotonin 5-$HT_{2C}$; $G_{12/13}$ for adenosine 2A ($A_{2A}$) and prostaglandin E1 ($EP_1$) receptors; $G_{i/o}$ for corticotropin-releasing hormone receptor 1 (CRFR1), $ET_A$ and G protein-coupled receptor 39 (GPR39))

---

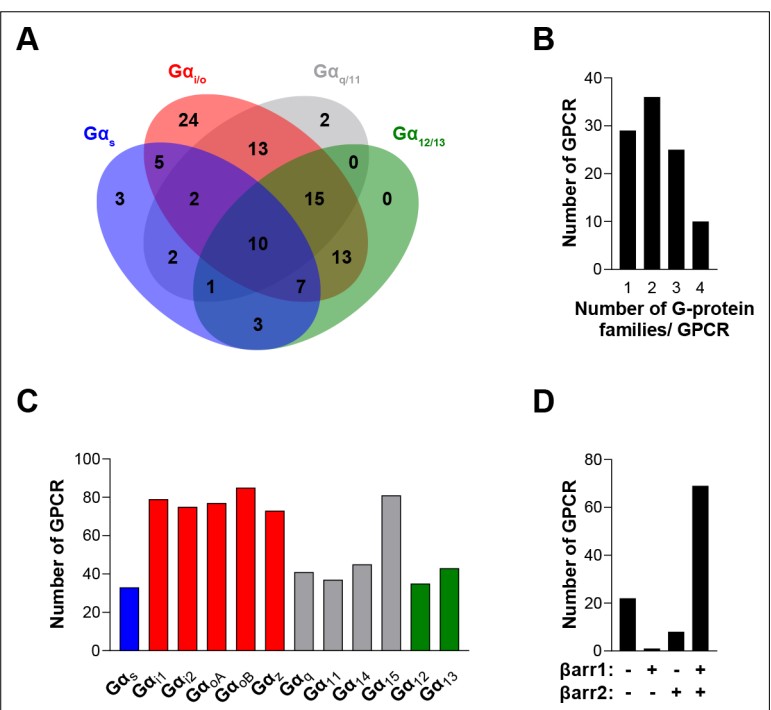

**Figure 4.** The EMTA ebBRET platform has a unique ability to uncover coupling selectivity between G protein families. (**A**) Venn diagram showing the numbers of receptors coupled to each G protein family in the EMTA ebBRET biosensor assay. (**B**) Evaluation of receptors coupling promiscuity: number of receptors that couple to members of 1, 2, 3, or 4 G protein families. (**C**) Determination of G protein subunit coupling frequency: number of receptors that activate each Gα subunit. (**D**) Proportion of receptors recruiting βarrestins: number of receptors that do not recruit (-/-) or that recruit either (+/- or -/+) or both (+/+) βarrestin isotypes. All data are based on double normalized $E_{max}$ values from *Figure 3*.

The online version of this article includes the following source data and figure supplement(s) for figure 4:

**Source data 1.** Raw data of *Figure 4*.

**Figure supplement 1.** G protein subtypes distribution across the 100 GPCRs profiled with the EMTA ebBRET-based platform.

**Figure supplement 1—source data 1.** Raw data of *Figure 4—figure supplement 1*.

or higher (ex: $G_z$ for serotonin 5-HT$_{2B}$; $G_{15}$ for adenosine 3 (A$_3$) and melanocortin 3 (MC3R) receptors; $G_{12}$ for bradykinin 2 (B$_2$), cholecystokinin A (CCK$_1$), chemokine receptor 6 (CCR6) and ET$_A$ receptors; $G_{12/13}$ for CRFR1 and GPR68) than those for the canonical ones. Interestingly, in many instances the potency for the newly uncovered couplings are similar to those for βarrestins, which is generally lower than for their canonical G proteins, a finding consistent with the role of βarrestins in signaling arrest at the plasma membrane. The potency differences observed for the activation of different G protein subtypes by a given receptor may lead to preferential activation of some pathways over others. This relative selectivity is likely to be influenced by tissue-dependent G protein subtype expression levels. The physiological consequences of such selectivity remain to be investigated.

When examining the frequency of coupling for each Gα subunit family (*Figure 4C*), the $G_{i/o}$ family members were the most commonly activated, with 89% of the tested receptors activating a $G_{i/o}$ family member. In contrast, only 33%, 49%, and 45% of the receptors activate $G_s$, $G_{12/13}$, or $G_{q/11}$ (excluding Gα$_{15}$) family members, respectively. Not surprisingly, and consistent with its reported promiscuous coupling, Gα$_{15}$ was found to be activated by 81% of the receptors. For some receptors, we also observed preferential coupling of distinct members within a subtype family (*Figure 3—figure supplement 1*). For instance, 33% of $G_{i/o}$-coupled receptors can couple to only a subpopulation of the family (*Figure 4—figure supplement 1A*). For the $G_{q/11}$ family, only 44% activate all family members with 45% activating only Gα$_{15}$ and 11% engaging only two or three members of the family. A matrix expressing the % of receptors engaging a specific Gα subtype that also activated another subtype, is illustrated in *Figure 4—figure supplement 1B*. When considering individual families, considerable variation within the $G_{i/o}$ family was observed. The greatest similarities were observed between Gα$_{oB}$ and either Gα$_{oA}$ or Gα$_z$, and the lowest between Gα$_{i1}$ and Gα$_z$. A striking example of intra-family coupling selectivity is the serotonin 5-HT$_{2B}$ that activates only Gα$_{oB}$ and Gα$_z$ and GPR65 that selectively activates Gα$_{oB}$. Similarly, when considering the ligand-promoted responses above our threshold criteria (see Materials and methods), histamine H$_2$ and MC3R receptors show preferred activation of Gα$_{oB}$ and Gα$_z$, whereas the prostaglandin F (FP) and neuropeptide Y5 (Y$_5$) receptors preferentially activate Gα$_{oB}$, Gα$_{oA}$, and Gα$_z$. Even when all members of a given family are found to be activated, some receptors activate specific family members with greater potencies (*Supplementary file 2C*).

When considering βarrestin recruitment, our analysis shows that 22% of receptors did not recruit βarrestin 1 or 2, even in the presence of overexpressed GRK2 (*Figure 4D*). Among the receptors able to recruit βarrestins, only a very small number selectively recruited βarrestin1 (1.3%) or βarrestin2 (6.4%), most of them recruiting both βarrestins in the presence of GRK2 (92.3%) (*Figure 4D*). Over-expression of GRK2 potentiated the recruitment of βarrestin2 for 68% of receptors highlighting the importance of GRK2 expression level in determining βarrestin activation (*Supplementary files 3 and 2*).

## Comparison with previous datasets reveals commonalities and crucial differences

We compared the signaling profiles obtained here with those presented by *Inoue et al., 2019* and the GtP dataset. Of note, this comparison only considers the final reported couplings that in the Inoue's study were based on the criteria of positive coupling if LogRAi $\geq$ –1 and negative coupling if LogRAi $\leq$ –1, and is influenced by the different cut-offs and normalization used in the two studies. A comparison of couplings using common E$_{max}$ standard deviation cut-off, quantitative normalization and aggregation of G proteins into families is provided in the accompanying paper (*Hauser et al., 2022*). As can be seen in *Supplementary file 4A*, among the 70 receptors common to both studies, less couplings were detected in our study than reported in Inoue et al. for Gα$_s$ (21 *vs.* 28), Gα$_{i1}$ (54 *vs.* 56), Gα$_q$ (31 *vs.* 34), and Gα$_{14}$ (36 *vs.* 40). In contrast, more receptors activating Gα$_{12}$ (29 *vs.* 23), Gα$_o$ (59 *vs.* 41), Gα$_{13}$ (30 *vs.* 15), Gα$_z$ (52 *vs.* 37), and Gα$_{15}$ (62 *vs.* 15) were detected in our study. When comparing with data collected in GtP, that reports couplings grouped for G protein families (*i.e.*: $G_s$, $G_{i/o}$, $G_{q/11}$, or $G_{12/13}$) and not at the single G protein subtype level, we detected less couplings than what was reported in GtP for Gα$_s$ (32 *vs.* 37), but more for Gα$_{i/o}$ (89 *vs.* 69), Gα$_{q/11}$ (81 *vs.* 48), and Gα$_{12/13}$ (47 *vs.* 10), among the 99 receptors common to both datasets (*Supplementary file 4B*).

Altogether, the comparative analysis reveals 64% and 69% identity of couplings between the EMTA and Inoue's or GtP datasets, respectively. Each dataset reporting unique couplings and missing couplings found in the other two datasets. The reasons for these differences are plausibly due to

intrinsic differences in the assays used. For instance, for $G_{12/13}$ and $G_{15}$ specifically, the difference with the GtP dataset most likely results from the fact that in most cases $G_{12/13}$ or $G_{15}$ activation were determined indirectly since, until their recent description ($G_{12/13}$: *Quoyer et al., 2013*; *Schrage et al., 2015*; $G_{15}$:*Inoue et al., 2019*; *Olsen et al., 2020*), no robust readily available assay existed to monitor the activation of these G proteins.

## Validation of newly identified $G_{12/13}$ and $G_{15}$ couplings

Given the overrepresentation of both $G_{12/13}$ and $G_{15}$ couplings, obtained with the EMTA assays *vs.* those reported by Inoue et al. and the GtP datasets, the validity of the EMTA assay to detect real productive couplings, was confirmed using orthogonal assays for selective examples not reported in the two other datasets. For $G_{12/13}$, we used the PKN-based BRET biosensor detecting Rho activation downstream of either $G_{12/13}$ or $G_{q/11}$ (*Namkung et al., 2018*) and the MyrPB-Ezrin-based BRET biosensor detecting the activation of Ezrin downstream of $G_{12/13}$ (*Leguay et al., 2021*), both in the absence of heterologously expressed G proteins. Ligand stimulation of FP and CysLT$_2$ receptors led to Rho and ezrin activation (*Figure 3—figure supplement 3A*), that were insensitive to the $G_{q/11}$ inhibitor YM-254890, confirming that these receptors activate $G\alpha_{12/13}$.

For newly identified $G_{15}$ couplings, we took advantage of the lack of $G\alpha_{15}$ in HEK293 cells and assessed the impact of $G\alpha_{15}$ heterologous expression on receptor-mediated calcium responses (*Figure 3—figure supplement 3B*). For prostaglandin E2 (EP$_2$) and κ-opioid (κOR) receptors, which couple to $G_{15}$ but no other $G_{q/11}$ members, expression of $G\alpha_{15}$ significantly increased the PGE2- and Dynorphin A- promoted calcium responses. For $\alpha_{2A}$ adrenergic ($\alpha_{2A}$AR) and vasopressin 2 (V$_2$) receptors that couple other $G_{q/11}$ family members, treatment with YM-254890 completely abolished the agonist-promoted calcium response in the absence of $G\alpha_{15}$. In contrast, the calcium response evoked by $\alpha_{2A}$AR and V$_2$ agonists following $G\alpha_{15}$ expression was completely insensitive to YM-254890 (*Figure 3—figure supplement 3B*), confirming that these receptors can activate this YM-254890-insensitive G protein subtype (*Takasaki et al., 2004*).

## EMTA platform detects constitutive receptor activity and biased signaling

We went on to assess the ability of the EMTA platform to detect receptor constitutive activity. Transfection of increasing amounts of adenosine A$_1$ receptor (A$_1$) led to a receptor-dependent increase in basal ebBRET of the $G\alpha_{i2}$-activation sensor (*Figure 5A*, left), reflecting A$_1$ constitutive activity. The A$_1$ inverse agonist DPCPX (*Lu et al., 2014*) dose-dependently decreased the constitutive A$_1$-mediated activation of $G\alpha_{i2}$ (*Figure 5A*, left), indicating that EMTA can detect inverse agonism. Although we can not exclude that the high basal activity resulted from activation by adenosine in the cell culture medium, the fact that high basal activity was observed for A$_1$ but not A$_3$, despite a similar potency of adenosine to activate these two receptors subtypes (see *Figure 5—figure supplement 1A*), supports the notion that the increased basal activity reflects A$_1$ constitutive activity.

To further confirm that the platform can adequately detect inverse agonism, a second receptor for which no endogenous ligand should be present in the media, the CB$_1$ receptor, was used. As illustrated in *Figure 5A* (right), increase CB$_1$ expression led to a ligand-independent constitutive activation of G$_z$, that could be completely blocked by the CB$_1$ inverse agonist rimonabant.

EMTA also faithfully detected biased signaling. Indeed, as previously reported (*Namkung et al., 2018*; *Wei et al., 2003*), angiotensin analogs such as SII, saralasin or TRV027 displayed biased signaling by promoting efficient βarrestin2 recruitment but marginal or no $G\alpha_q$, $G\alpha_{i2}$, or $G\alpha_{13}$ activation as compared to angiotensin II that activated all G proteins and βarrestin2 (*Figure 5B*). The platform was also used to identify biased signaling resulting from single nucleotide polymorphisms. As shown in *Figure 5C*, two naturally occurring variants of human GPR17 (isoform 2) localised in the TM3 E/DRY motif resulted in altered functional selectivity profiles. Whereas the Asp128Asn variant displayed WT-like activity on $G\alpha_{i2}$, it lost the ability to activate $G\alpha_q$ and βarrestin2. In contrast, variant Arg129His at the neighboring position resulted in an increased constitutive βarrestin2 recruitment and a loss of $G\alpha_{i2}$ and $G\alpha_q$ protein signaling.

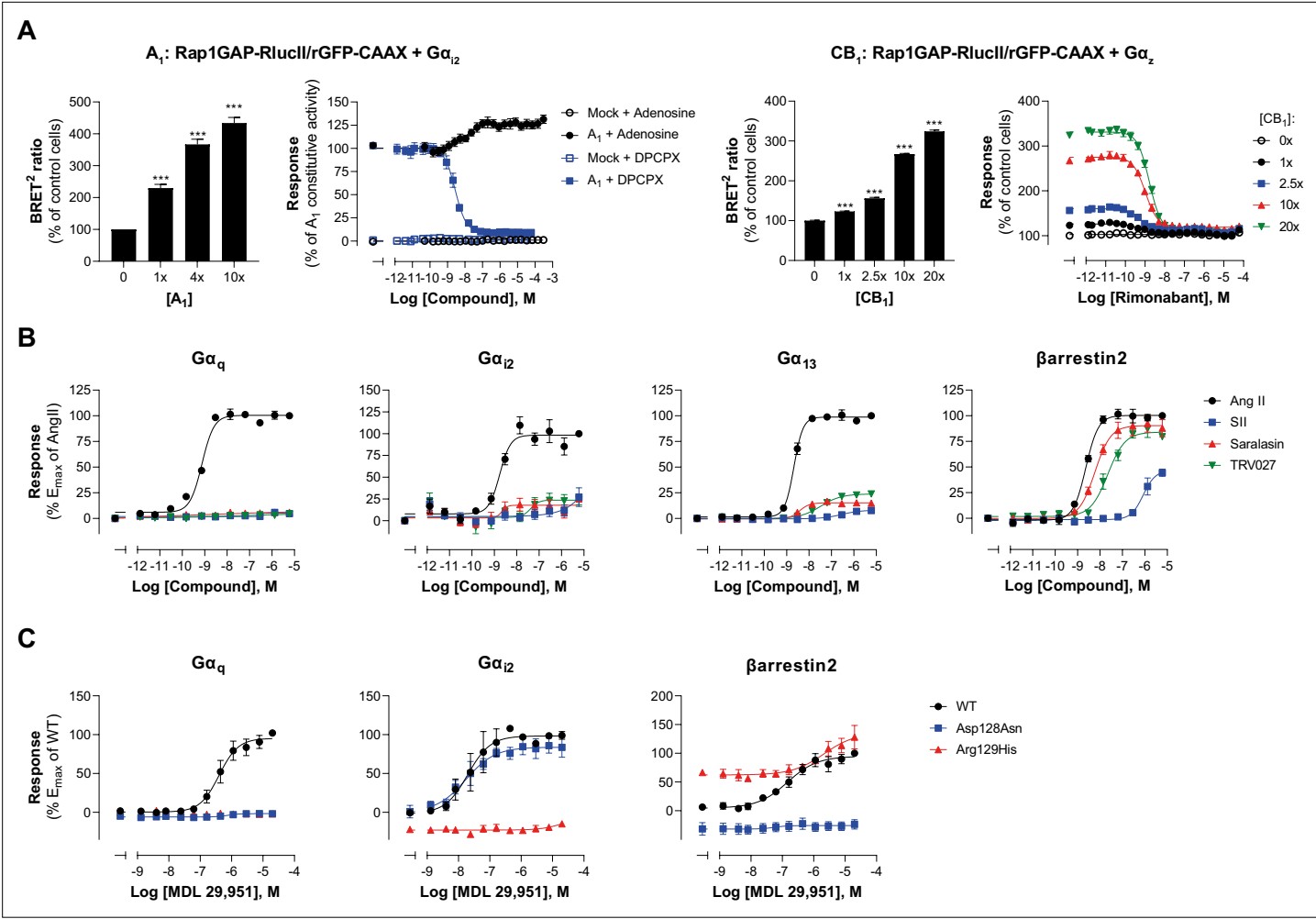

**Figure 5.** Multiple applications using the EMTA ebBRET platform. (**A**) Inverse agonist activity detection. *Left*: $G\alpha_{i2}$ activation in HEK293 cells transfected with the Rap1GAP-RlucII/rGFP-CAAX sensors with untagged $G\alpha_{i2}$ and increasing amount of $A_1$ receptor plasmid. Data are expressed in % of response obtained in control cells (0 ng of $A_1$) and are the mean ± SEM of 4–6 independent experiments performed in two replicates. One Way ANOVA test: ***$p < 0.001$ compared to control cells. HEK293 cells expressing the $G\alpha_{i2}$ activation sensor and control (Mock) or $A_1$ receptor plasmid were stimulated (10 min) with increasing concentrations of the indicated compound. Data are expressed in % of constitutive response obtained in vehicle-treated $A_1$ transfected cells and are the mean ± SEM of 4-6 independent experiments performed in one replicate. *Right*: $G\alpha_z$ activation in HEK293 cells transfected with the Rap1GAP-RlucII/rGFP-CAAX sensors with untagged $G\alpha_z$ and increasing amount of $CB_1$ receptor plasmid. Data are expressed in % of response obtained in control cells (0 ng of $CB_1$) and are the mean ± SEM of 4 independent experiments performed in one replicate. One Way ANOVA test: ***$p < 0.001$ compared to control cells. HEK293 cells expressing the $G\alpha_z$ activation sensor and increasing amount of $CB_1$ receptor plasmid were directly stimulated (10 min) with increasing concentrations of the $CB_1$ inverse agonist rimonabant. Data are expressed as % of the response obtained in control cells (0 ng of $CB_1$) treated with vehicle and are the mean ± SEM of 4 independent experiments performed in one replicate. (**B**) Ligand-biased detection. Concentration-response curves of $AT_1$ for the endogenous ligand (Angiotensin II, AngII) and biased agonists [Sar1-Ile4-Ile8] AngII (SII), saralasin or TRV027. G protein and βarrestin2 signaling activity were assessed by EMTA platform. Data are expressed in % of maximal response elicited by AngII and are the mean ± SEM of 3–6 independent experiments performed in one replicate. (**C**) Functional selectivity of naturally occurring receptor variants. Concentration-response curves for WT or E/DRY motif Asp128Asn and Arg129His variants of GPR17 upon agonist stimulation in HEK293 cells co-expressing the indicated EMTA biosensor. Data are expressed in % of maximal response elicited by WT receptor and are the mean ± SEM of 3 independent experiments performed in one replicate.

The online version of this article includes the following source data and figure supplement(s) for figure 5:

**Source data 1.** Raw data of *Figure 5*.

**Figure supplement 1.** Modulation of ligand-promoted response detected by EMTA ebBRET platform by receptor constitutive activity.

**Figure supplement 1—source data 1.** Raw data of *Figure 5—figure supplement 1*.

# Combining $G_z$ and $G_{15}$ biosensors for safety panels and systems pharmacology

The G protein coupling profiles obtained for the 100 GPCRs revealed that 95% of receptors activate either $G\alpha_z$ (73%) or $G\alpha_{15}$ (81%). Measuring activation of both pathways simultaneously provides an almost universal sensor applicable to screening. Combining the two sensors (Rap1GAP-RlucII/p63-RhoGEF-RlucII/rGFP-CAAX) in the same cells allowed to detect ligand concentration-dependent activation of a safety panel of 24 GPCRs, that are well established as contributors to clinical adverse

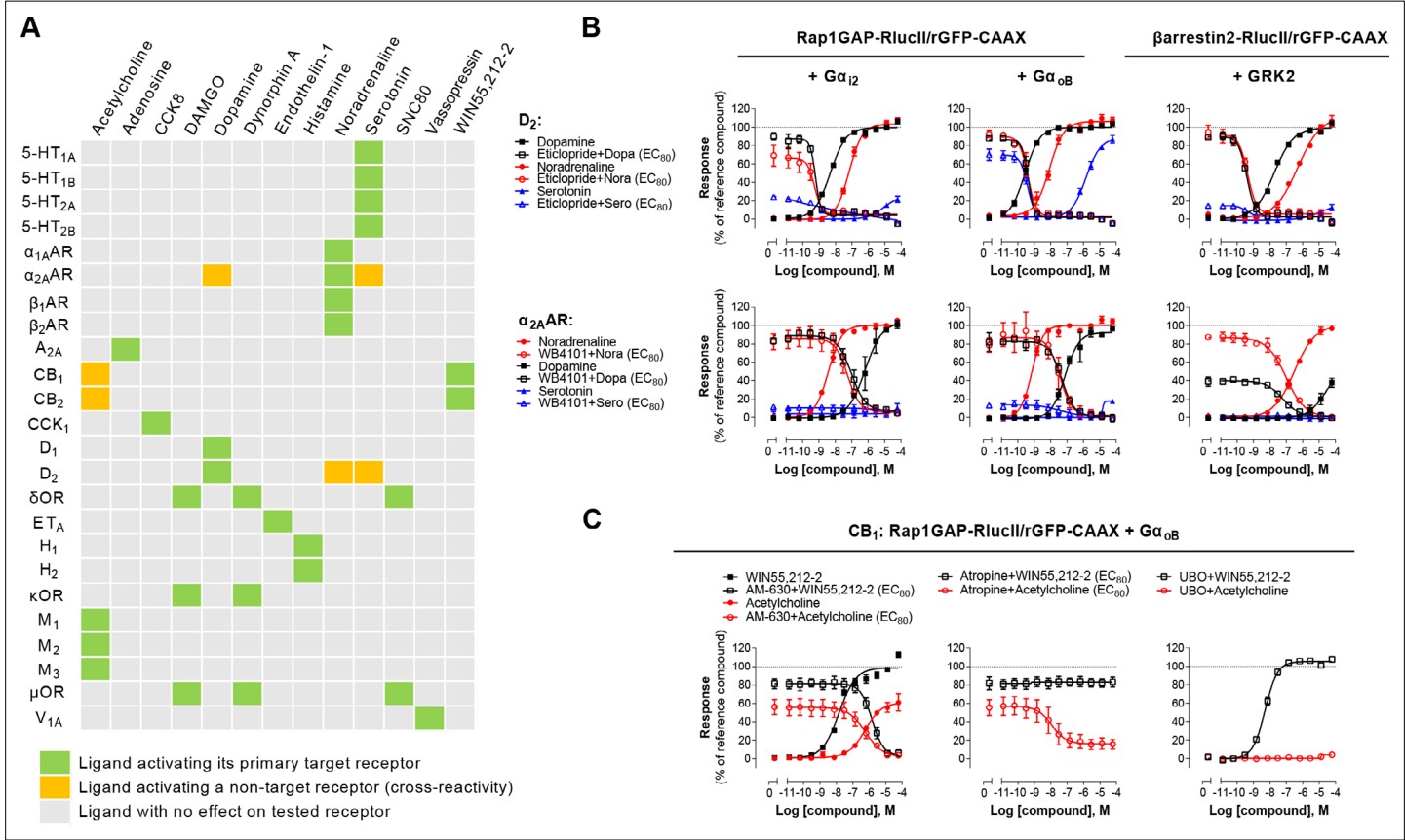

**Figure 6.** Detection of direct and indirect (*trans*) mechanisms of ligand polypharmacology using the $G_z/G_{15}$ biosensor. (**A**) Test of the $G_z/G_{15}$ biosensor on a safety target panel. ebBRET signal was measured before and after stimulation with the indicated ligand in HEK293 cells transfected with the combined $G_z/G_{15}$ biosensor and one of the 24 receptors listed. (**B**) Cross-activation of $D_2$ and $\alpha_{2A}AR$ by other natural ligands. For the agonist mode read, HEK293 cells expressing $D_2$ or $\alpha_{2A}AR$ and either the $G\alpha_{i2}$, $G\alpha_{oB}$, or the βarrestin2 + GRK2 sensors were stimulated with increasing concentrations of the indicated ligand. For the antagonist mode read, cells were pretreated with increasing concentrations of the selective $D_2$ antagonist eticlopride or the selective $\alpha_{2A}AR$ antagonist WB4101 before stimulation with an $EC_{80}$ of the indicated ligand. Data are the mean ± SEM from 3-4 independent experiments performed in one replicate and expressed in % of the response elicited by dopamine or noradrenaline for $D_2$ and $\alpha_{2A}AR$ expressing cells, respectively. (**C**) Indirect (*trans*) activation of $CB_1$ by acetylcholine. For the agonist mode read, HEK293 cells expressing $CB_1$ and the Rap1GAP-RlucII/rGFP-CAAX sensors with untagged $G\alpha_{oB}$ were stimulated with increasing concentrations of the indicated ligand. For the antagonist mode read, same cells were pretreated or not with increasing concentrations of the CB inverse agonist AM-630 (*left*) or the cholinergic antagonist atropine (*central*) before stimulation with an $EC_{80}$ of the indicated ligand. To evaluate the contribution of $G_{q/11}$-coupled receptor, cells were pretreated with the $G\alpha_q$ inhibitor UBO-QIC and then stimulated with increasing concentrations of the indicated ligand (*right*). Data are the mean ± SEM from 3-5 independent experiments performed in one replicate and expressed in % of the response elicited by WIN55,212–2.

The online version of this article includes the following source data and figure supplement(s) for figure 6:

**Source data 1.** Raw data of *Figure 6*.

**Figure supplement 1.** Combined $G_z/G_{15}$ biosensor.

**Figure supplement 1—source data 1.** Raw data of *Figure 6—figure supplement 1*.

**Figure supplement 2.** Validation of direct activation of $\alpha_{2A}AR$ by dopamine.

**Figure supplement 2—source data 1.** Raw data of *Figure 6—figure supplement 2*.

drug reactions (*Bowes et al., 2012*; *Figure 6—figure supplement 1*). Indeed, the $G_z/G_{15}$ sensor captured the activation of receptors largely or uniquely coupled to either $G\alpha_z$ (e.g. $CB_2$) or $G\alpha_{15}$ (e.g. $A_{2A}$ and $A_{2B}$), as well as receptors coupled (to varying degrees) to both pathways. The usefulness of the $G_z/G_{15}$ combined sensor to detect off-target ligand activity is illustrated in *Figure 6A*. Most ligands tested were specific for their primary target(s). However, certain ligands displayed functional cross-reactivity with GPCRs other than their cognate targets. These included the activation of the $\alpha_{2A}AR$ by dopamine and serotonin, the $D_2$ by noradrenaline and serotonin, and of the $CB_1$ and $CB_2$ receptors by acetylcholine (*Figure 6B–C*). The activation of $D_2$ by noradrenaline and serotonin was confirmed by the ability of the $D_2$-family selective antagonist eticlopride to block the dopamine-, serotonin-, and noradrenaline-promoted responses detected using the combined $G_z/G_{15}$ or the $G_{i2}$- and $G_{oB}$-selective sensors and βarrestin2 sensor (*Figure 6B*, top). Similarly, use of the $\alpha_{2A}AR$ selective antagonist, WB4101, allowed to confirm that dopamine can activate $G\alpha_{i2}$, $G\alpha_{oB}$ and βarrestin2 through the $\alpha_{2A}AR$ (*Figure 6B*, bottom). Such pleiotropic activation of different monoaminergic receptors by catecholamines and serotonin has been previously observed (*Roth et al., 2004*; *Sánchez-Soto et al., 2016*; *Sunahara et al., 1991*). Direct activation of the $\alpha_{2A}AR$ by dopamine was confirmed by showing that treatment with the $D_2$-family receptor selective antagonist eticlopride had negligible effect on dopamine-mediated activation of $G\alpha_{i2}$ and $G\alpha_{oB}$ in cells heterologously expressing $\alpha_{2A}AR$, confirming that the response did not result from the activation of endogenously expressed dopamine receptor. In contrast, eticlopride blocked the activation of $G\alpha_{i2}$ and $G\alpha_{oB}$ in cells heterologously expressing $D_2$ (*Figure 6—figure supplement 2*).

These cross-reactivity may be direct (i.e. via direct binding of a ligand to its non-cognate receptor) as suggested above, or indirect (e.g. 'trans', via ligand activation of its canonical receptor, leading to subsequent secretion of factors that activate the non-canonical target). One such example of trans-activation is provided by the activation of cannabinoid $CB_1$ and $CB_2$ receptors by acetylcholine (detected by the $G_{z/15}$ and confirmed with the $G_{oB}$ sensors; *Figure 6A and C*). Indeed, the activation was completely inhibited by both the CB inverse agonist AM-630 and by the cholinergic antagonist atropine (*Figure 6C*, left). Yet the response evoked by the CB selective agonist WIN55,212 2 was not blocked by atropine (*Figure 6C*, center). $G\alpha_{oB}$ activation by acetylcholine did not result from direct activation of endogenous muscarinic receptors since no $G\alpha_{oB}$ response was observed in parental cells (*Figure 3—figure supplement 2*). Given that the $M_3$ muscarinic receptor, which is endogenously expressed at relatively high levels in HEK293 cells (*Atwood et al., 2011*), is strongly coupled to the $G_{q/11}$, $CB_1$-expressing cells were pretreated with $G_{q/11/14}$ inhibitor UBO-QIC prior to stimulation with acetylcholine. UBO-QIC pre-treatment blocked acetylcholine- but not WIN55,212–2-mediated $G\alpha_{oB}$ activation (*Figure 6C*, right). These results demonstrate that $CB_1$ activation by acetylcholine is indirect and potentially involves the secretion of an endogenous CBR ligand following activation of $G_{q/11}$ by endogenous muscarinic acetylcholine receptors. The combined $G_z/G_{15}$ sensor is therefore a useful tool to identify interplay between receptors and to explore systems pharmacology resulting from such cross-talks.

## Discussion

This study describes the development and validation of a genetically encoded ebBRET-based biosensor platform allowing live-cell mapping of GPCR-G protein coupling preferences covering 12 heterotrimeric G proteins. The novel EMTA biosensors were combined with previously described ebBRET-based βarrestin trafficking sensors (*Namkung et al., 2016*), providing an unprecedented description of GPCR signaling partner couplings. In addition to providing a resource to study GPCR functional selectivity (*Pándy-Szekeres et al., 2022*) , the sensors provide versatile and readily usable tools to study, on a large-scale, pharmacological processes such as constitutive activity, inverse agonism, ligand-biased signaling, and signaling cross-talk.

Our EMTA-based biosensor platform offers several advantages relative to other available approaches. First, EMTA provides direct real-time measurement of proximal signaling events following GPCR activation (i.e. Gα protein activation and βarrestin recruitment) and resulting in lower level of amplification than those of assays relying on enzymatic activity of downstream effectors (i.e.: adenylyl cyclase or phospholipase C) or artificial detection systems (i.e.: gene-reporter or TGF-α shedding assays) that measure signal accumulation sometimes following extended incubation times. In addition,

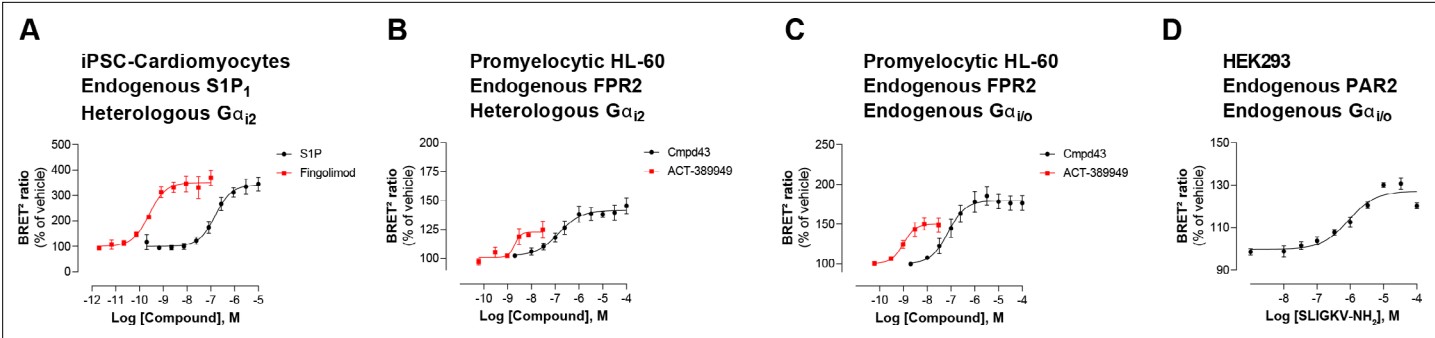

**Figure 7.** Detection of endogenous receptor- and/or G protein-mediated responses in cells with the EMTA ebBRET platform. Concentration-dependent activation of $G\alpha_{i2}$ protein by (**A**) endogenous $S1P_1$ receptor in iPSC-derived cardiomyocytes transfected with heterologous $G\alpha_{i2}$, (**B**) endogenous FPR2 in promyelocytic HL-60 cells transfected with heterologous $G\alpha_{i2}$, (**C**) endogenous FPR2 in promyelocytic HL-60 cells with endogenous $G_{i/o}$ proteins and (**D**) endogenous PAR2 receptor in HEK293 cells with endogenous $G_{i/o}$ proteins. In all cases, cells were co-transfected with the Rap1GAP-RlucII/rGFP-CAAX biosensor. Data are the mean ± SEM of 3-4 independent experiments performed in one replicate and are expressed as BRET$^2$ ratio in percentage of response induced by vehicle.

The online version of this article includes the following source data for figure 7:

**Source data 1.** Raw data of *Figure 7*.

measuring proximal activity reduces the risk of cross-talks between pathways that may complicate data interpretation when considering downstream signaling as the readout (*Mancini et al., 2015*).

Second, EMTA uses native untagged GPCRs and G protein subunits (except for $G_s$), contrary to protein complementation (*Laschet et al., 2019*), FRET/BRET-based $G\alpha\beta\gamma$ dissociation/receptor-G protein interaction (*Bünemann et al., 2003*; *Galés et al., 2005*; *Galés et al., 2006*; *Hoffmann et al., 2005*; *Namkung et al., 2018*; *Olsen et al., 2020*) or TGF-α shedding (*Inoue et al., 2019*) assays. Modifying these core-signaling components could alter responses, complicate interpretation and explain some of the discrepancies observed between the EMTA platform and other approaches used to study G protein activation. Moreover, the ability to work with unmodified receptors and G proteins (except for $G_s$) offers numerous advantages. First, it allows for the detection of endogenous GPCR signaling in either generic HEK293 cells (*Figure 3—figure supplement 2*) or more physiologically relevant cell lines such as induced pluripotent stem cell (iPSC)-derived cardiomyocytes (*Figure 7A*) and promyelocytic HL-60 cells (*Figure 7B*). Further it allows, in cells expressing sufficient endogenous level of the G proteins of interest, to detect activation of both native receptor and G proteins with no need of overexpression (*Figure 7C–D*). This is illustrated by the ability to detect the recruitment of Rap1GAP upon activation of the endogenous $G_{i/o}$ family members by the formyl peptide receptor 2 (FPR2) in HL-60 cells (*Figure 7C*) or protease-activated receptor-2 (PAR2) in HEK293 cells (*Figure 7D*). The ability to detect the activation of endogenous G protein was also illustrated in *Figure 2—figure supplement 1I*, where the responses elicited by agonist stimulation were lost in cells genetically deleted of the G protein engaged by the studied receptor (i.e.: $G_{q/11}$ or $G_{i/o}$ families). Recently, another BRET-based approach (*Maziarz et al., 2020*), taking advantage of a synthetic peptide recognizing the GTP-bound form of $G\alpha$ subunits, also allows the detection of native G protein activation, offering alternative means to probe coupling selectivity profiles for both endogenously and heterologously expressed GPCRs.

Finally, similarly to BERKY, the EMTA assay platform detects the active form of the $G\alpha$ subunits rather than the surrogate measurement of $G\alpha\beta\gamma$ dissociation (*Galés et al., 2005*; *Masuho et al., 2015*; *Maziarz et al., 2020*; *Mende et al., 2018*), which can also detect non-productive binding as recently described for the $V_2$ engagement of $G_{12}$ (*Okashah et al., 2020*).

A potential caveat of EMTA is the use of common downstream effectors for all members of a given G protein family. Indeed, one cannot exclude that distinct members of a given family may display different relative affinities for their common effector. However, such differences are compensated by our data normalization that establishes the maximal response observed for a given subtype as the reference for this pathway (*Figure 3A*), as long as the number of the diversity of receptors included in the analysis is sufficient.

A second potential caveat of EMTA is that, when using heterologously expressed GPCRs and G proteins, some of the responses could result from favorable stoichiometries that may not exist under physiological conditions. It follows that such profiling represents the coupling possibilities of a given GPCR and not necessarily the coupling that will be observed in all cell types. Any couplings observed in such high-throughput studies requires further validation to conclude on their physiological relevance in cells or tissues of interest, and to form hypothesis for futures studies. Because we elected to use unmodified receptors (i.e.: not bearing any tags), the expression level of receptors could not be directly monitored. However, the double normalization method developed (see Materials and methods) allows quantitative comparison of coupling preferences across different receptors curtailing the influence of the assay response windows as well as receptor expression levels. Indeed, the double normalization allows ranking the coupling propensity of the receptors first as a function of the receptor which shows the strongest coupling to a specific G protein subtype, and then establishing the maximal response observed for a given G protein subtype as the reference for all G protein activated by a given receptor. In addition, as illustrated using the $ET_A$ receptor as example, titrating receptor levels did not influence the $pEC_{50}$ for the activation of the different G protein coupled to this receptor (*Figure 2—figure supplement 3B* and *Supplementary file 1C*). Similarly, the $pEC_{50}$ was not affected when titrating the amount of G protein subtype expressed (*Figure 2—figure supplement 3A* and *Supplementary file 1B*). As expected, only the amplitude of the response was affected.

It could be argued that overexpressing the G protein effectors (i.e.: p63-RhoGEF, Rap1GAP or PDZ-RhoGEF) used as sensors could influence the couplings observed. This potential caveat is mitigated by the fact that we used truncated part and/or modified versions of these effectors that limit the possibilities of interference with other components of the signaling machinery, and served essentially as a binding detector of the active forms of the G proteins (see Materials and methods). Supporting this notion, titrating the amount of the biosensor effector component did not affect the $pEC_{50}$ of G protein activation (*Figure 2—figure supplement 3C* and *Supplementary file 1D*).

Another limitation of the EMTA platform is the lack of a soluble effector protein selective for activated $G\alpha_s$ thus requiring tagging of the $G\alpha_s$ subunit (*Figure 1B*, bottom) and monitoring its dissociation from the plasma membrane. Yet, our data show that this translocation reflects $G_s$ activation state, justifying its use in a G protein activation detection platform.

Finally, because EMTA is able to detect constitutive activity, high receptor expression levels may lead to an elevated basal signal level that may obscure an agonist-promoted response. Such an example can be appreciated for the $A_1$ receptor for which the agonist-promoted $G\alpha_{i2}$ response did not reach the activation threshold criteria because of a very high constitutive activity level (*Figure 5A*). The impact of receptor expression on the constitutive activity and the narrowing on the agonist-promoted response is illustrated for $G\alpha_q$ activation by the 5-$HT_{2C}$ (*Figure 5—figure supplement 1B*).

A limitation of any large-scale signaling study and drug discovery program is that ligands may elicit responses downstream of receptors other than the one under study. The development of a $G_z/G_{15}$ quasi-universal biosensor enables efficient screening and detection of such polypharmacology and cross-talk. Using a combination of EMTA and appropriate pharmacological tools, we also proposed a systematic approach to distinguish off-target action of ligands from cross-talk. Interestingly, the cross-talk between the $M_3$ and CB receptors detected (*Figure 6*) may have physiological relevance since activation of muscarinic acetylcholine receptors has been shown to enhance the release of endocannabinoids in the hippocampus (*Kim et al., 2002*). The combined $G_z/G_{15}$ biosensor should be particularly useful for early profiling of compound activity on safety panels and for the design of drugs displaying polypharmacology, an approach that is increasingly considered for the development of neuropsychiatric drugs (*Roth et al., 2004*).

The EMTA platform undoubtedly represents a novel tool-set that could be amenable for high throughput screening of small molecules and biologics across an array of signaling pathways, allowing for the discovery of functionally selective molecules or for GPCR deorphanization campaigns. The ability of the EMTA platform to quantitatively assess G protein coupling selectivity firmly expands the concept of functional selectivity and potential ligand bias beyond the dichotomic G protein *vs.* βarrestin view and provides plausible functional selectivity profiles that could be tested for their biological and pharmacological outcomes.

# Materials and methods

## Cells

HEK293 clonal cell line (HEK293SL cells), hereafter referred as HEK293 cells, were a gift from S. Laporte (McGill University, Montreal, Quebec, Canada) and previously described (*Namkung et al., 2016*). HEK293 cells devoid of functional $G\alpha_s$ ($\Delta G_s$), $G\alpha_{12}$ and $G\alpha_{13}$ ($\Delta G_{12/13}$), $G\alpha_q$, $G\alpha_{11}$, $G\alpha_{14}$ and $G\alpha_{15}$ ($\Delta G_{q/11}$) and, $G\alpha_i$, and $G\alpha_o$ ($\Delta G_{i/o}$) proteins were a gift from Dr. A. Inoue (Tohoku University, Sendai, Miyagi, Japan) and previously described (*Devost et al., 2017*; *Namkung et al., 2018*; *Schrage et al., 2015*; *Stallaert et al., 2017*). Cells were maintained in Dulbecco's Modified Eagle Medium (DMEM, Wisent, Saint-Jean-Baptiste, QC, Canada) supplemented with 10% fetal bovine serum (FBS, Wisent) and 1% antibiotics (100 U/mL penicillin and 100 µg/mL streptomycin (PS); Wisent). HL-60 cells were obtained from ATCC and maintained in RPMI 1640 medium containing L-Glutamine and 25 mM HEPES (Gibco) supplemented with 20% FBS (Wisent) and 1/100 volume PS (Wisent). Differentiation of HL-60 cells into neutrophil-like cells was induced by maintaining the cells in growth medium containing 1.3% DMSO (Bioshop) during 5 days. Cardiomyocytes derived from induced pluripotent stem cells (iPSCs; iCell Cardiomyocytes) were obtained from FUJIFILM Cellular Dynamics (Madison, WI, USA) and maintained in maintenance medium provided with the cells (special formulation by FujiFilm). Cells were grown at 37 °C in 5% $CO_2$ and 90% humidity and checked for mycoplasma contamination.

## Plasmids and ebBRET biosensor constructs

Only human GPCRs and human $G\alpha$ subunits were used in this study. An open reading frame of each full-length GPCR was cloned into pcDNA3.1(+) expression plasmid. Except when otherwise specified, GPCRs sequences were devoid of epitope tags.

$G\alpha_s$-67-RlucII (*Carr et al., 2014*), $G\alpha_{i1}$-loop-RlucII and GFP10-$G\gamma_1$ (*Armando et al., 2014*), $G\alpha_{i2}$-loop-RlucII and βarrestin2-RlucII (*Quoyer et al., 2013*), $G\alpha_{oB}$-99-RlucII (*Mende et al., 2018*), $G\alpha_q$-118-RlucII (*Breton et al., 2010*), $G\alpha_{12}$-136-RlucII and PKN-RBD-RlucII (*Namkung et al., 2018*), $G\alpha_{13}$-130-RlucII (*Avet et al., 2020*), GFP10-$G\gamma_2$ (*Galés et al., 2006*), βarrestin1-RlucII (*Zimmerman et al., 2012*), rGFP-CAAX (*Namkung et al., 2016*), EPAC (*Leduc et al., 2009*), MyrPB-Ezrin-RlucII (*Leguay et al., 2021*), HA-$β_2$AR (*Lavoie et al., 2002*), signal peptide-Flag-$AT_1$ (*Goupil et al., 2015*), and EAAC-1 (*Brabet et al., 1998*) were previously described. Full-length, untagged $G\alpha$ subunits, $G\beta_1$ and $G\gamma_9$ were purchased from cDNA Resource Center. GRK2 was generously provided by Dr. Antonio De Blasi (Istituto Neurologico Mediterraneo Neuromed, Pozzilli, Italy).

To selectively detect $G_{i/o}$ activation, a construct coding for aa 1–442 of Rap1 GTPase-activating protein (comprising a $G_{i/o}$ binding domain) fused to Rluc8, was sequence-optimized, synthetized and subcloned at TopGenetech (St-Laurent, QC, Canada). From this construct, a RlucII-tagged version of Rap1GAP (1-442) with a linker sequence (GSAGTGGRAIDIKLPAT) between Rap1GAP and RlucII was created by Gibson assembly in pCDNA3.1_Hygro (+) GFP10-RlucII, replacing GFP10. Three substitutions (i.e. S437A/S439A/S441A) were introduced into the Rap1GAP sequence by PCR-mediated mutagenesis. These putative (S437 and S439) and documented (S441) (*McAvoy et al., 2009*) protein kinase A phosphorylation sites were removed in order to eliminate any $G_s$-mediated Rap1GAP recruitment to the plasma-membrane.

To selectively detect $G_{q/11}$ activation, a construct encoding the $G_q$ binding domain of the human p63 Rho guanine nucleotide exchange factor (p63RhoGEF; residues: 295–502) tagged with RlucII was done from IMAGE clones (OpenBiosystems; Burlington, ON, Canada) and subcloned by Gibson assembly in pCDNA3.1_Hygro (+) GFP10-RlucII, replacing GFP10. The $G_q$ binding domain of p63RhoGEF and RlucII were separated by the peptidic linker ASGSAGTGGRAIDIKLPAT. N-term part containing palmitoylation sites maintaining p63 to plasma membrane and part of its DH domain involved in RhoA binding/activation (*Aittaleb et al., 2010*; *Aittaleb et al., 2011*) are absent of the sensor.

To selectively detect $G_{12/13}$ activation, a construct encoding the $G_{12/13}$ binding domain of the human PDZ-RhoGEF (residues: 281–483) tagged with RlucII was done by PCR amplification from IMAGE clones (OpenBiosystems) and subcloned by Gibson assembly in pCDNA3.1_Hygro (+) GFP10-RlucII, replacing GFP10. The peptidic linker GIRLREALKLPAT is present between RlucII and the $G_{12/13}$ binding domain of PDZ-RhoGEF. The sensor is lacking the PDZ domain of PDZ-RhoGEF involved in protein-protein interaction, as well as actin-binding domain and DH/PH domains involved in GEF activity and RhoA activation (*Aittaleb et al., 2010*).

The sequence of each EMTA biosensors is provided in the *Supplementary file 5*.

## Transfection

For BRET experiments, HEK293 cells (1.2 mL at $3.5 \times 10^5$ cells per mL) were transfected with a fixed final amount of pre-mixed biosensor-encoding DNA (0.57 µg, adjusted with salmon sperm DNA; Invitrogen) and human receptor DNA. Transfections were performed using a polyethylenimine solution (PEI, 1 mg/mL; Polysciences, Warrington, PA, USA) diluted in NaCl (150 mM, pH 7.0; 3:1 PEI/DNA ratio). Gelatin solution (1%; Sigma-Aldrich, Saint-Louis, Missouri) was used to stabilize DNA/PEI transfection mixes. Following addition of cells to the stabilized DNA/PEI transfection mix, cells were immediately seeded ($3.5 \times 10^4$ cells/well) into 96-well white microplates (Greiner Bio-one; Monroe, NC, USA) and maintained in culture for the next 48 hr in DMEM containing 2% FBS and 1% PS. DMEM medium without L-glutamine (Wisent) was used for transfection of cells with mGluR to avoid receptor activation and desensitization. For Neutrophil-like differentiated HL-60 cells, cells were resuspended in electroporation medium (growth medium containing an extra 15 mM of HEPES pH 7.0) at $25 \times 10^6$ cells/mL. Electroporation reactions were prepared by adding 50 µL of DNA mastermix (20 µg total of DNA adjusted with salmon sperm DNA, supplemented with 210 mM NaCl) to 200 µL of cell suspension and transferring into 0.4 cm gap electroporation cuvettes (Bio-Rad). The cells were electroporated at 350 µF/400 V using a Bio-Rad Gene Pulser II electroporation system, washed in electroporation medium, and seeded in 96-well plates at $0.8 \times 10^6$ cells/well in 200 µL of growth medium. BRET assays were performed 6 hr post-electroporation. For iPSC Cardiomyocytes, cells were seeded in 96-well plates pretreated with fibronectin (10 µg/ml 60 min; Sigma-Aldrich) at $3.5 \times 10^4$ cells /well. After 48 hr, attached iPSCs cells were transfected with the indicated biosensor components, using TransIT-LT1 reagent (Mirus; Madison, WI, USA), according to manufacturer recommendation. BRET assays were performed 48 hr after transfection.

For $Ca^{2+}$ experiments, cells ($3.5 \times 10^4$ cells/well) were co-transfected with the indicated receptor, with or without $G\alpha_{15}$ protein, using PEI and seeded in poly-ornithine-coated 96-well clear-bottom black microplates (Greiner Bio-one) and maintained in culture for the next 48 hr.

For BRET-based imagery, cells ($4 \times 10^5$ cells/dish) were seeded into 35 mm poly-d-lysine-coated glass-bottom culture dishes (Mattek Corporation; Ashland, MA, USA) in 2 ml of fresh medium and incubated at 37 °C in 5% $CO_2$, 3 day before imaging experiments. Twenty-four hours later, cells were transfected with EMTA ebBRET biosensors and the indicated receptor (i.e. p63-RhoGEF-RlucII/rGFP-CAAX + $G\alpha_q$ and $AT_1$, Rap1GAP-RlucII/rGFP-CAAX + $G\alpha_{i2}$ and $D_2$ or PDZ-RhoGEF-RlucII/rGFP-CAAX + $G\alpha_{13}$ and TP$\alpha$R) using X-tremeGENE 9 DNA transfection reagent (3:1 reagent/DNA ratio; Roche) diluted in OptiMEM (Gibco) and maintained in culture for the next 48 hr in DMEM containing 10% FBS and 1% PS.

## Bioluminescence resonance energy transfer measurement

Enhanced bystander BRET (ebBRET) was used to monitor the activation of each G$\alpha$ protein, as well as $\beta$arrestin 1 and 2 recruitment to the plasma membrane. $G\alpha_s$ protein activation was measured between the plasma membrane marker rGFP-CAAX and human $G\alpha_s$-RlucII in the presence of human $G\beta_1$, $G\gamma_9$ and the tested receptor. $G\alpha_s$ downstream cAMP production was determined using the EPAC biosensor and GPBA receptor. $G\alpha_{i/o}$ protein family activation was followed using the selective-$G_{i/o}$ effector Rap1GAP-RlucII and rGFP-CAAX along with the human $G\alpha_{i1}$, $G\alpha_{i2}$, $G\alpha_{oA}$, $G\alpha_{oB}$, or $G\alpha_z$ subunits and the tested receptor. $G\alpha_{q/11}$ protein family activation was determined using the selective-$G_{q/11}$ effector p63-RhoGEF-RlucII and rGFP-CAAX along with the human $G\alpha_q$, $G\alpha_{11}$, $G\alpha_{14}$, or $G\alpha_{15/16}$ subunits and the tested receptor. $G\alpha_{12/13}$ protein family activation was monitored using the selective-$G_{12/13}$ effector PDZ-RhoGEF-RlucII and rGFP-CAAX in the presence of either $G\alpha_{12}$ or $G\alpha_{13}$ and the tested receptor. The expression level of the G$\alpha$ subunits was monitored by western blot in HEK293 cells that endogenously expressed $G\alpha_{i1}$, $G\alpha_{i2}$, $G\alpha_{12}$, $G\alpha_{13}$, $G\alpha_q$, $G\alpha_{11}$, $G\alpha_{14}$, and G$\alpha$s but not $G\alpha_{oA}$, $G\alpha_{oB}$, $G\alpha_z$, and $G\alpha_{15}$ (*Figure 2—figure supplement 6*). $G\alpha_{12/13}$-downstream activation of the Rho pathway was measured using PKN-RBD-RlucII or Ezrin-RlucII and rGFP-CAAX with the indicated receptor. $\beta$arrestin recruitment to the plasma membrane was determined using DNA mix containing rGFP-CAAX and $\beta$arrestin1-RlucII with GRK2 or $\beta$arrestin2-RlucII alone or with GRK2 and the tested receptor. Glutamate transporters EAAC-1 and EAAT-1 were systematically co-transfected with the mGluR to prevent receptor activation and desensitization by glutamate secreted in the medium by the cells (*Brabet et al., 1998*). All ligands were also tested for potential activation of endogenous receptors by transfecting the biosensors without receptor DNA. The $G_z/G_{15}$ biosensor consists of a combination of the

following plasmids: rGFP-CAAX, Rap1GAP-RlucII, $G\alpha_z$, p63-RhoGEF-RlucII and $G\alpha_{15}$. For G protein activation detection using the BRET-based $G\alpha\beta\gamma$ dissociation sensors, cells were co-transfected with untagged $G\beta_1$ and $G\alpha_q$-118-RlucII, $G\alpha_{12}$-136-RlucII or $G\alpha_{13}$-130-RlucII with GFP10-$G\gamma_1$, or $G\alpha_{i1}$-loop-RlucII, $G\alpha_{i2}$-loop-RlucII or $G\alpha_{oB}$-99-RlucII with GFP10-$G\gamma_2$, along with the indicated receptor.

The day of the BRET experiment, cells were incubated in HBSS for 1 hr at room temperature (RT). Cells were then co-treated with increasing concentrations of ligand (see Appendix 1—key resources table and *Supplementary file 2* for details) and the luciferase substrate coelenterazine prolume purple (1 µM, NanoLight Technologies; Pinetop, AZ, USA) for 10 min at RT. Plates were read on a Synergy Neo microplate reader (BioTek Instruments, Inc; Winooski, VT, USA) equipped with 410 ± 80 nm donor and 515 ± 30 nm acceptor filters or with a Spark microplate reader (Tecan; Männedorf, Switzerland) using the BRET$^2$ manufacturer settings. The BRET signal (BRET$^2$) was determined by calculating the ratio of the light intensity emitted by the acceptor over the light intensity emitted by the donor. To validate the specificity of the biosensor responses, cells were pretreated in the absence or presence of either the $G\alpha_q$ inhibitor UBO-QIC (100 nM, 30 min; Institute for Pharmaceutical Biology of the University of Bonn, Germany), the $G\alpha_{i/o}$ inhibitor PTX (100 ng/mL, 18 hr; List Biological Laboratories, Campbell, California, USA) or the $G\alpha_s$ activator CTX (0–200 ng/mL, 4 hr; Sigma-Aldrich) before stimulation with agonist. For inverse agonist activity detection of $A_1$ or $CB_1$ receptors, cells were stimulated during 10 min with increasing concentrations of DPCPX or rimonabant, respectively. For ligand-cross receptor activation experiments, cells were pretreated for 10 min with increasing concentrations of antagonists or inverse agonist (eticlopride for $D_2$, WB4101 for $\alpha_{2A}$AR, atropine for muscarinic receptors and AM-630 for $CB_1$) before a 10 min stimulation with an $EC_{80}$ concentration of the indicated agonist. BRET was measured as described above. For the safety target panel ligand screen using the combined $G_z/G_{15}$ sensor, basal ebBRET level was first measured 10 min following the addition of coelenterazine prolume purple (1 µM) and ebBRET level was measured again following a 10 min stimulation with a single dose of the indicated ligand (1 µM for endothelin-1 and 10 µM for all other ligands). Technical replicates for each receptor were included on the same 96-well plate. For kinetics experiment of $ET_A$ activation, basal BRET was measured during 150 s before cells stimulation with either vehicle (DMSO) or 1 µM of endothelin-1 (at time 0 sec) and BRET signal was recorded each 30 s during 3570 s. For the validation of $G_{12/13}$-mediated signal by new identified $G_{12/13}$-coupled receptor using PKN- or Ezrin-based BRET biosensors, cells were pretreated or not with the $G\alpha_q$ inhibitor YM-254890 (1 µM, 30 min; Wako Pure Chemical Industries (Fujifilm), Osaka, Japan) before agonist stimulation for 10 min. For G protein activation detection using the BRET-based $G\alpha\beta\gamma$ dissociation sensors, and for titration experiments of either $G\alpha$ proteins subunit with GEMTA sensors, GPCRs with GEMTA sensors or Effector-RlucII (p63-RhoGEF-RlucII for $G\alpha_{q/11}$, Rap1GAP-RlucII for $G\alpha_{i/o}$ or PDZ-RhoGEF-RlucII for $G\alpha_{12/13}$) from GEMTA sensors, cells were stimulated with increasing concentrations of the indicated agonist in the presence of prolume purple for 10 min before BRET measurement. For BRET in iPSC cardiomyocytes and HL-60 cells, cells were incubated in Tyrode Hepes buffer (137 mM NaCl, 0.9 mM KCl, 1 mM MgCl$_2$, 11.9 mM NaHCO$_3$, 3.6 mM NaH$_2$PO$_4$, 25 mM HEPES, 5.5 mM D-Glucose and 1 mM CaCl$_2$, pH 7.4) 30 min at RT before being treated with increasing concentrations of agonist for 15 min, using prolume purple (2 µM) as luciferase substrate, and BRET measured.

## BRET data analyses and coupling efficiency evaluation

All BRET ratios were standardized using the equation below and represented as universal BRET (*u*BRET) values: *u*BRET = ((BRET ratio – A)/(B-A)) * 10,000. Constants A and B correspond to the following values:

A = pre-established BRET ratio obtained from transfection of negative control (vector coding for RlucII alone).

B = pre-established BRET ratio obtained from transfection of positive control (vector coding for a GFP10-RlucII fusion protein).

For a given signaling pathway, *u*BRET values at each agonist concentration were normalized as the % of the response obtained in the absence of agonist (vehicle) and concentration-response curves were fitted in GraphPad Prism 8.3 software using a four-parameter logistic nonlinear regression model. Results are expressed as mean ± SEM of at least three independent experiments.

A ligand-promoted response was considered real when the $E_{max}$ value was ≥to the mean + 2*SD of the response obtained in vehicle condition and that a $pEC_{50}$ value could be determined in the agonist

concentration range used to stimulate the receptor. Consequently, a score of 0 or 1 was assigned to each signaling pathway depending on an agonist's ability to activate the tested pathway (0 = no activation; 1 = activation). In the case were responses associated to endogenous receptor were detectable, we considered as 'distorted' and exclude all the responses observed in the presence of transfected receptor for which $E_{max}$ was ≤to 2*mean of the $E_{max}$ value obtained with endogenous receptors or $pEC_{50}$ was ≥to 2*mean of the $pEC_{50}$ value obtained with endogenous receptors. Consequently, a score of 0 was assigned for these distorted responses in radial graph representation (*Figure 3—figure supplement 1*) and concentration-response curves were placed on a gray background in signaling signature profile panels (*Supplementary file 3*). Whenever transfected receptors produced an increase in $E_{max}$ or a left-shift in $pEC_{50}$ values compared to endogenous receptors, responses were considered 'true' and were assigned with a score of 1 for radial graph representation (*Figure 3—figure supplement 1*) and concentration-response curves were placed on a yellow background in signaling signature profile panels to indicate a partial contribution of endogenous receptors (*Supplementary file 3*).

We used a double normalization of $E_{max}$ and $pEC_{50}$ values to compare the signaling efficiency obtained for the 100 GPCRs across all receptors and pathways. $E_{max}$ and $pEC_{50}$ values deduced from concentration-response curves were first normalized between 0 and 1 across receptors by ranking the receptors as a function of the receptor that most efficiently activate a given pathway and then using the activation value for the pathway (including G protein and βarrestin subtypes) that a given receptor most efficiently activate as a reference for the other pathways that can be activated by this receptor. This double normalization can be translated in the following formalized equation:

- STEP1: For each receptor and for each pathway:

$$\left[\frac{E_{max}\ GPCR_x}{E_{max}\ GPCR_{Ref}}\right]_{Pathway\ A} = \textit{Pathway specific normalized score for GPCR}_x \textit{ on pathway A ([PSNS GPCR}_x]_{Pathway\ A})$$

where: $GPCR_x$ is receptor being analyzed, $GPCR_{Ref}$ is the receptor giving greatest $E_{max}$ on pathway A of all receptors studied (i.e. reference receptor for pathway A). A PSNS was determined for every receptor and every pathway coupled to that receptor.

- STEP2: For any given receptor:

$$\frac{\left[PSNS\ GPCR_x\right]_{Pathway\ A}}{\left[PSNS\ GPCR_x\right]_{Ref\ pathway}} = \textit{Normalized pathway A coupling score for GPCR}_x$$

where: [PSNS GPCR$_x$] $_{Pathway\ A}$ is the pathway specific normalized score for GPCR$_x$ on pathway A, and [PSNS GPCR$_x$] $_{Ref\ pathway}$ is the pathway specific normalized score for the pathway giving the highest PSNS for GPCR$_x$ (i.e., reference pathway for GPCR$_x$).

For the safety target panel ligand screen using the combined $G_z/G_{15}$ sensor, the fold ligand-induced stimulation was calculated for each receptor by dividing the BRET ratio after ligand addition (measured at 10 min post stimulation) by the basal BRET ratio prior to receptor stimulation. Activation thresholds were defined as the mean + 2*SD of the ligand-stimulated response obtained with receptor-null cells expressing only the combined $G_z/G_{15}$ sensor.

## Ca²⁺ mobilization assay

The day of the experiment, cells were incubated with 100 µL of a $Ca^{2+}$-sensitive dye-loading buffer (FLIPR calcium five assay kit, Molecular Devices; Sunnyvale, CA, USA) containing 2.5 mM probenecid (Sigma-Aldrich) for 1 hr at 37 °C in a 5% $CO_2$ incubator. During a data run, cells in individual wells were exposed to an $EC_{80}$ concentration of agonist, and fluorescent signals were recorded every 1.5 s for 3 min using the FlexStation II microplate reader (Molecular Devices). For receptors that also activate other $G_{q/11}$ family members, cells were pretreated with the $G_{q/11}$ inhibitor YM-254890 (1 µM, 30 min) before agonist stimulation. $G\alpha_{15}$ is resistant to inhibition by YM-254890, thus allowing to measure $Ca^{2+}$ responses generated specifically by $G\alpha_{15}$.

## BRET-based imaging

BRET images were obtained as previously described (*Kobayashi et al., 2019*). Briefly, the day of imaging experiment, cells were carefully rinsed with HBSS, and images were acquired before and after agonists addition (100 nM for Angiotensin II and U46619, and 1 µM for dopamine) diluted in HBSS in the presence of the luciferase substrate coelenterazine prolume purple (20 µM).

Images were recorded using an inverted microscope (Nikon Eclipse Ti-U) equipped with x60 objective lens (Nikon CFI Apochromat TIRF) and EM-CCD camera (Nuvu HNu 512). Measurements were carried out in photon counting mode with EM gain 3000. Exposure time of each photon counting was 100ms. Successive 100 frames were acquired alternatively with 480 nm longpass filter (acceptor frames) or without filter (total luminescence frames), and integrated. Image integrations were repeated 10 times and 1000 frames (*Video 1*) or 5 times and 500 frames (*Videos 2 and 3*) of acceptor and total luminescence were used to generate each image.

BRET values were obtained by dividing acceptor counts by total luminescence counts pixelwise. BRET values from 0.0 to 0.8 (*Video 1*) or 0.0–0.5 (*Videos 2 and 3*) were allocated to 'jet' heatmap array using MATLAB 2019b. Brightness of each pixel was mapped from the signal level of total luminescence image. 0% and 99.9% signal strength were allocated to the lowest and highest brightness to exclude the influence of defective pixels with gamma correction factor of 2.0.

The movies were generated using ImageJ 1.52 a. Frame rate is 10 (*Video 1*) or 3 (*Videos 2 and 3*) frames/s, and frame interval is 20 or 100 s for *Videos 1 and 2–3*, respectively. The field of view of the movie is 137 µm x 137 µm.

## Western blot analysis

Cells were transfected or not with the indicated biosensors mix as previously described and whole-cell extracts were prepared 48 hr later. Briefly, cells were washed with ice-cold PBS and lysed in a buffer containing 10 mM Tris buffer (pH 7.4), 100 mM NaCl, 1 mM EDTA, 1 mM EGTA, 0.1% SDS, 1% Triton X-100, 10% Glycerol supplemented with protease inhibitors cocktails (Thermo Fisher Scientific). Cell lysates were centrifuged at 13,000 × g for 30 min at 4 °C. Equal amounts of proteins were separated by SDS-PAGE and transferred onto polyvinylidene fluoride membrane. The membranes were blocked (1 hr incubation at RT in TBS, 0.1% Tween-20, 5% BSA) and successively probed with primary antibody and appropriate goat secondary antibodies coupled to horseradish peroxidase (see **Appendix 1-Key Resources Table**). Western blots were visualized using enhanced chemiluminescence and detection was performed using a ChemiDoc MP Imaging System (BioRad). Relative densitometry analysis on protein bands was performed using MultiGauge software (Fujifilm). Results were normalized against control bands.

## Statistical analyses

Curve fitting and statistical analyses were performed using GraphPad Prism 8.3 software and methods are described in the legends of the figures. Significance was determined as $p < 0.05$.

# Acknowledgements

We thank Shane C Wright for scientific discussion and Monique Lagacé for critical reading of the manuscript. The authors are grateful to the funding from Bristol-Myers Squibb that supported the detection of Gα proteins by endogenous receptors in iPSC cardiomyocytes and promyelocytic HL-60 cells.

# Additional information

### Competing interests

Arturo Mancini, Claire Normand, Florence Gross, Sandra Morissette: was employee of Domain Therapeutics North America during this research. Billy Breton: was employee of Domain Therapeutics North America during this research. Has filed patent application (US20200256869A1) related to the biosensors used in this work and the technology has been licensed to Domain Therapeutics. Christian Le Gouill, Hiroyuki Kobayashi, Mireille Hogue, Viktoriya Lukasheva: has filed patent application (US20200256869A1) related to the biosensors used in this work and the technology has been licensed to Domain Therapeutics. Eric B Fauman, Jean-Philippe Fortin: is employee and shares holders of Pfizer. Stephan Schann, Xavier Leroy: is employee and is part of the management of Domain Therapeutics. Michel Bouvier: is the president of Domain Therapeutics scientific advisory board. Has filed patent

application (US20200256869A1) related to the biosensors used in this work and the technology has been licensed to Domain Therapeutics. The other authors declare that no competing interests exist.

## Funding

| Funder | Grant reference number | Author |
|---|---|---|
| Canada Research Chairs | | Michel Bouvier |
| Canadian Institutes of Health Research | FDN-148431 | Michel Bouvier |
| Lundbeckfonden | R218-2016-1266 | David E Gloriam |
| Lundbeckfonden | R313-2019-526 | David E Gloriam |
| Novo Nordisk Fonden | NNF18OC0031226 | David E Gloriam |
| Lundbeckfonden | R278-2018-180 | Alexander S Hauser |
| Bristol-Myers Squibb | | Michel Bouvier |

The funders had no role in study design, data collection and interpretation, or the decision to submit the work for publication.

## Author contributions

Charlotte Avet, Conceptualization, Formal analysis, Investigation, Methodology, Writing – original draft, Writing – review and editing; Arturo Mancini, Conceptualization, Formal analysis, Investigation, Methodology, Resources, Writing – original draft, Writing – review and editing; Billy Breton, Conceptualization, Investigation, Methodology; Christian Le Gouill, Conceptualization, Methodology; Alexander S Hauser, Formal analysis, Writing – review and editing; Claire Normand, Hiroyuki Kobayashi, Florence Gross, Mireille Hogue, Viktoriya Lukasheva, Stéphane St-Onge, Marilyn Carrier, Sandra Morissette, Investigation; Madeleine Héroux, Investigation, Supervision; Eric B Fauman, Jean-Philippe Fortin, Resources; Stephan Schann, Funding acquisition, Resources; Xavier Leroy, Conceptualization, Resources, Supervision; David E Gloriam, Formal analysis, Funding acquisition, Supervision, Writing – original draft, Writing – review and editing; Michel Bouvier, Conceptualization, Formal analysis, Funding acquisition, Methodology, Resources, Supervision, Writing – original draft, Writing – review and editing

## Author ORCIDs

Alexander S Hauser ![ORCID] http://orcid.org/0000-0003-1098-6419
Hiroyuki Kobayashi ![ORCID] http://orcid.org/0000-0003-4965-0883
David E Gloriam ![ORCID] http://orcid.org/0000-0002-4299-7561
Michel Bouvier ![ORCID] http://orcid.org/0000-0003-1128-0100

## Decision letter and Author response

Decision letter https://doi.org/10.7554/eLife.74101.sa1
Author response https://doi.org/10.7554/eLife.74101.sa2

# Additional files

## Supplementary files

• Supplementary file 1. Impact of G proteins, receptor or effector-RlucII titration on absolute pEC50 values. (A) Absolute $pEC_{50}$ values of responses elicited in WT *vs.* Knockout Gα protein background cells. $pEC_{50}$ values deduced from dose-response curves for various receptor tested in parental (WT) HEK293 cells or devoid of $G_s$, $G_{12/13}$, $G_{q/11}$ or $G_{i/o}$ proteins are related to *Figure 2—figure supplement 1*. (B) Absolute $pEC_{50}$ values of responses elicited in cells transfected with different amounts of Gα proteins. $pEC_{50}$ values deduced from dose-response curves obtained following Gα subunit titration in HEK293 cells transfected with GEMTA sensors and related to *Figure 2—figure supplement 3A*. (C) Absolute $pEC_{50}$ values of responses elicited in cells transfected with different amounts of receptors. $pEC_{50}$ values deduced from dose-response curves obtained following $ET_A$ titration in HEK293 cells transfected with GEMTA sensors and related to *Figure 2—figure supplement 3B*. (D) Absolute $pEC_{50}$ values of responses elicited in cells transfected with different amounts of Effector-RlucII. $pEC_{50}$ values deduced from dose-response curves obtained following

Effector-RlucII titration in HEK293 cells transfected with GEMTA sensors and related to *Figure 2—figure supplement 3C*.

• Supplementary file 2. List of tested receptors and ligands, along with the raw $E_{max}$, absolute $pEC_{50}$ and their corresponding double normalized (dnor) values. The $E_{max}$ (in % of vehicle response) and absolute $pEC_{50}$ values deduced from concentration-response curves for the 100 GPCRs tested as well as the double normalized $E_{max}$ and $pEC_{50}$ values calculated are related to *Supplementary file 3* and *Figure 3*, respectively.

• Supplementary file 3. Signaling profiles of 100 therapeutically-relevant human GPCRs using the EMTA ebBRET platform. Concentration-response curves in HEK293 cells expressing the indicated biosensor after stimulation of heterologously expressed receptor with the indicated ligand. Data are the mean ± SEM from at least 3 independent experiments and expressed in % of the response obtained for cells treated with vehicle. For ligands that elicited endogenous receptor-mediated responses (curves with light gray and yellow background for responses similar to and better responses than those obtained with the endogenous receptors, respectively), curves from cells expressing endogenous or heterologously expressed receptors are shown in *Figure 3—figure supplement 2*.

• Supplementary file 4. Comparison of G protein couplings identified with EMTA platform and other datasets. Comparison of G protein couplings identified with EMTA platform and TGF-α shedding assay in *Inoue et al., 2019* (A) or reported in GtP database (B).

• Supplementary file 5. EMTA biosensor amino acid sequences.

• Transparent reporting form

### Data availability

All data generated or analysed during this study are included in the manuscript and supporting file; Source Data files have been provided for Figures 2, 4, 5, 6 and 7 and associated figure supplements. Supplementary file 1 contains the numerical data used to generate Figure 2-figure supplement 1 and Figure 2-figure supplement 3. Supplementary File 2 contains the numerical data used to generate figure 3. Further data are available in the companion paper co-published in eLife: https://doi.org/10.7554/eLife.74107.

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

# Appendix 1.

## Key Resources Table

**Appendix 1—key resources table**

| Reagent type (species) or resource | Designation | Source or reference | Identifiers | Additional information |
|---|---|---|---|---|
| Cell line (*Homo-sapiens*) | HEK293 | 10.1038/ncomms12178 (**Namkung et al., 2016**) | | HEK293 clonal cell line (HEK293SL cells) |
| Cell line (*Homo-sapiens*) | $\Delta G_s$ HEK293 cells | Dr. A. Inoue (Tohoku University, Sendai, Miyagi, Japan) 10.1124 / mol.116.106419 (**Stallaert et al., 2017**) | | HEK293 cells devoid of functional $G\alpha_s$ |
| Cell line (*Homo-sapiens*) | $\Delta G_{12/13}$ HEK293 cells | Dr. A. Inoue (Tohoku University, Sendai, Miyagi, Japan) 10.1074/jbc.M116.763854 (**Devost et al., 2017**) | | HEK293 cells devoid of functional $G\alpha_{12}$ and $G\alpha_{13}$ |
| Cell line (*Homo-sapiens*) | $\Delta G_{q/11}$ HEK293 cells | Dr. A. Inoue (Tohoku University, Sendai, Miyagi, Japan) 10.1038/ncomms10156 (**Schrage et al., 2015**) | | HEK293 cells devoid of functional $G\alpha_q$, $G\alpha_{11}$, $G\alpha_{14}$ and $G\alpha_{15}$ |
| Cell line (*Homo-sapiens*) | $\Delta G_{I/o}$ HEK293 cells | Dr. A. Inoue (Tohoku University, Sendai, Miyagi, Japan) | | HEK293 cells devoid of functional $G\alpha_i$ and $G\alpha_o$ |
| Cell line (*Homo-sapiens*) | HL-60 | ATCC | Cat. #: CCL-240 | |
| Cell line (*Homo-sapiens*) | iCell Cardiomyocytes, 01434 | FUJIFILM Cellular Dynamics | Cat. #: R1057 | |
| Transfected construct (*Homo sapiens*) | Human Gα subunits-encoding plasmid library | Missouri S&T cDNA Resource Center (https://www.cdna.org/) | Cat. #: GNAI100000; GNAI200000; GNA0OA0000; GNA0OB0000; GNA0Z00000; GNA1200000; GNA1300001; GNA0Q00000; GNA1100000; GNA1400000; GNA1500000; GNA0SL0000 | |
| Transfected construct (*Homo sapiens*) | Gβ1 | Missouri S&T cDNA Resource Center (https://www.cdna.org/) | Cat. #: GNB0100000 | |
| Transfected construct (*Homo sapiens*) | Gγ9 | Missouri S&T cDNA Resource Center (https://www.cdna.org/) | Cat. #: GNG0900000 | |
| Transfected construct (*Homo sapiens*) | Gαs-67-RlucII | 10.1074/jbc.M114.618819(**Carr et al., 2014**) | | |
| Transfected construct (*Homo sapiens*) | Gαi1-loop-RlucII | 10.1096/fj.13–242446 (**Armando et al., 2014**) | | |
| Transfected construct (*Homo sapiens*) | Gαi2-loop-RlucII | 10.1073/pnas.1312515110 (**Quoyer et al., 2013**) | | |
| Transfected construct (*Homo sapiens*) | GαoB-99-RlucII | 10.1073/pnas.1804003115(**Mende et al., 2018**) | | |

*Appendix 1 Continued on next page*

*Appendix 1 Continued*

| Reagent type (species) or resource | Designation | Source or reference | Identifiers | Additional information |
|---|---|---|---|---|
| Transfected construct (*Homo sapiens*) | Gαq-118-RlucII | 10.1016 /j.bpj.2010.10.025 (***Breton et al., 2010***) | | |
| Transfected construct (*Homo sapiens*) | Gα12–136-RlucII | 10.1126/scisignal.aat1631(***Namkung et al., 2018***) | | |
| Transfected construct (*Homo sapiens*) | Gα13–130-RlucII | 10.1038 /s42003-020-01453-8 (***Avet et al., 2020***) | | |
| Transfected construct (*Homo sapiens*) | GFP10-Gγ1 | 10.1096/fj.13–242446 (***Armando et al., 2014***) | | |
| Transfected construct (*Homo sapiens*) | GFP10-Gγ2 | 10.1038/nsmb1134(***Galés et al., 2006***) | | |
| Transfected construct (*Homo sapiens*) | EPAC | 10.1124/jpet.109.156398(***Leduc et al., 2009***) | | |
| Transfected construct (*Homo sapiens*) | rGFP-CAAX | 10.1038/ncomms12178(***Namkung et al., 2016***) | | |
| Transfected construct (*Homo sapiens*) | Rap1GAP-RlucII | This paper | | See Materials and Methods |
| Transfected construct (*Homo sapiens*) | p63-RhoGEF-RlucII | This paper | | See Materials and Methods |
| Transfected construct (*Homo sapiens*) | PDZ-RhoGEF-RlucII | This paper | | See Materials and Methods |
| Transfected construct (*Homo sapiens*) | PKN-RBD-RlucII | 10.1126/scisignal.aat1631(***Namkung et al., 2018***) | | |
| Transfected construct (*Homo sapiens*) | MyrPB-Ezrin-RlucII | 10.1242/jcs.255307(***Leguay et al., 2021***) | | |
| Transfected construct (*Homo sapiens*) | βarrestin1-RlucII | 10.1126/scisignal.2002522(***Zimmerman et al., 2012***) | | |
| Transfected construct (*Homo sapiens*) | βarrestin2-RlucII | 10.1073/pnas.1312515110 (***Quoyer et al., 2013***) | | |
| Transfected construct (*Homo sapiens*) | GRK2 | This paper | | See Materials and Methods |
| Transfected construct (*Homo sapiens*) | RAMP3 | Domain Therapeutics North America | N/A | |

*Appendix 1 Continued on next page*

*Appendix 1 Continued*

| Reagent type (species) or resource | Designation | Source or reference | Identifiers | Additional information |
|---|---|---|---|---|
| Transfected construct (*Homo sapiens*) | EAAC-1 | 10.1016/s0028-3908(98)00091-4 (**Brabet et al., 1998**) | | |
| Transfected construct (*Homo sapiens*) | EAAT-1 | Domain Therapeutics North America | N/A | |
| Transfected construct (*Homo sapiens*) | signal peptide-Flag-AT$_1$ | 10.1074/jbc.M114.631119(**Goupil et al., 2015**) | | |
| Transfected construct (*Homo sapiens*) | FLAG-α$_{2B}$AR | Domain Therapeutics North America | N/A | |
| Transfected construct (*Homo sapiens*) | HA-β$_2$AR | 10.1074/jbc.M204163200(**Lavoie et al., 2002**) | | |
| Antibody | Gαi1 (I-20) (Rabbit polyclonal) | Santa Cruz | Cat. #: sc-391 RRID: AB_2247692 | WB (1:500) |
| Antibody | Gαi2 (T-19) (Rabbit polyclonal) | Santa Cruz | Cat. #: sc-7276 RRID:AB_2111472 | WB (1:500) |
| Antibody | Gαo (K-20) (Rabbit polyclonal) | Santa Cruz | Cat. #: sc-387 RRID:AB_2111641 | WB (1:500) |
| Antibody | Gαz (Rabbit monoclonal) | Abcam | Cat. #: ab154846 | WB (1:1000) |
| Antibody | Gαs (K-20) (Rabbit polyclonal) | Santa Cruz | Cat. #: sc-823 RRID:AB_631538 | WB (1:500) |
| Antibody | Gα12 (S-20) (Rabbit polyclonal) | Santa Cruz | Cat. #: sc-409 RRID:AB_2263416 | WB (1:500) |
| Antibody | Gα13 (A-20) (Rabbit polyclonal) | Santa Cruz | Cat. #: sc-410 RRID:AB_2279044 | WB (1:500) |
| Antibody | Gαq (E-17) (Rabbit polyclonal) | Santa Cruz | Cat. #: sc-393 RRID:AB_631536 | WB (1:500) |
| Antibody | Gα11 (C-terminal) (Rabbit polyclonal) | Sigma-Aldrich | Cat. #: SAB2109181 | WB (1:500) |
| Antibody | Gα14 (Rabbit polyclonal) | Sigma-Aldrich | Cat. #: SAB4300771 | WB (1:500) |
| Antibody | Gα15 (Rabbit polyclonal) | ThermoFisher scientific (Pierce) | Cat. #: PA1-29022 RRID:AB_1958024 | WB (1:5,000) |
| Antibody | βactin (Mouse monoclonal) | Sigma-Aldrich | Cat. #: A5441 RRID:AB_476744 | WB (1:5,000) |
| Antibody | Anti-rabbit HRP-coupled (Donkey polyclonal) | GE Healthcare | Cat. #: NA934 RRID:AB_772206 | WB (1:5,000) |
| Antibody | Anti-mouse HRP-coupled (Sheep polyclonal) | GE Healthcare | Cat. #: NA931 RRID:AB_772210 | WB (1:10,000) |
| Commercial assay or kit | FLIPR Calcium 5 Assay Kit | Molecular Devices | Cat. #: R8185 | |
| Chemical compound, drug | α-linolenic acid | Cayman Chemical | Cat. #: 21,910 | |
| Chemical compound, drug | α-MSH | Sigma-Aldrich | Cat. #: M4135 | |
| Chemical compound, drug | γ-MSH | Tocris | Cat. #: 4,272 | |

*Appendix 1 Continued on next page*

*Appendix 1 Continued*

| Reagent type (species) or resource | Designation | Source or reference | Identifiers | Additional information |
|---|---|---|---|---|
| Chemical compound, drug | [Pyr1]-Apelin 13 | Tocris | Cat. #: 2,420 | |
| Chemical compound, drug | [Sar1, Ile4,8]-Angiotensin II | Peptides International | Cat. #: PAN-4476-V-1EA | |
| Chemical compound, drug | 3-hydroxyoctanoic acid (3-HOA) | Sigma-Aldrich | Cat. #: H3898 | |
| Chemical compound, drug | 7α–25 dihydroxycholesterol | Sigma-Aldrich | Cat. #: SML0541 | |
| Chemical compound, drug | Acetylcholine chloride | Tocris | Cat. #: 2,809 | |
| Chemical compound, drug | ACT-389949 | Provided by Bristol-Myers Squibb | N/A | |
| Chemical compound, drug | Adenosine | Sigma-Aldrich | Cat. #: A9251 | |
| Chemical compound, drug | AM-630 | Tocris | Cat. #: 1,120 | |
| Chemical compound, drug | Amylin | Tocris | Cat. #: 3,418 | |
| Chemical compound, drug | Angiotensin II (Ang II) | Sigma-Aldrich | Cat. #: A9525 | |
| Chemical compound, drug | Arginine vasopressin (AVP) | Sigma-Aldrich | Cat. #: V9879 | |
| Chemical compound, drug | Atropine | Sigma-Aldrich | Cat. #: A0132 | |
| Chemical compound, drug | Bovine serum albumin | Sigma-Aldrich | Cat. #: A7030 | |
| Chemical compound, drug | C5a | Complement Technology | Cat. #: A144(300) | |
| Chemical compound, drug | Calcitonin | Bachem | Cat. #: H-2250 | |
| Chemical compound, drug | CCK Octapeptide, sulfated (CCK8) | Tocris | Cat. #: 1,166 | |
| Chemical compound, drug | CCL20 | R&D Systems | Cat. #: 360-MP/CF | |
| Chemical compound, drug | CCL3 (MIP-1a) | R&D Systems | Cat. #: 270-LD/CF | |
| Chemical compound, drug | Cholera Toxin (CTX) from *Vibrio cholerae* | Sigma-Aldrich | Cat. #: C8052 | |
| Chemical compound, drug | Cmpd43 | Provided by Bristol-Myers Squibb | N/A | |

*Appendix 1 Continued on next page*

*Appendix 1 Continued*

| Reagent type (species) or resource | Designation | Source or reference | Identifiers | Additional information |
|---|---|---|---|---|
| Chemical compound, drug | Corticotropin-Releasing Factor (CRF) | Bachem | Cat. #: H-2435 | |
| Chemical compound, drug | CXCL12 | R&D Systems | Cat. #: 350-NS | |
| Chemical compound, drug | CXCL13 | R&D Systems | Cat. #: 801 CX/CF | |
| Chemical compound, drug | CXCL8 | R&D Systems | Cat. #: 208-IL/CF | |
| Chemical compound, drug | DAMGO | Tocris | Cat. #: 1,171 | |
| Chemical compound, drug | Dopamine | Sigma-Aldrich | Cat. #: H8502 | |
| Chemical compound, drug | DPCPX | Tocris | Cat. #: 0439 | |
| Chemical compound, drug | Dynorphin A | Tocris | Cat. #: 3,195 | |
| Chemical compound, drug | Endothelin-1 | Tocris | Cat. #: 1,160 | |
| Chemical compound, drug | Eticlopride | Tocris | Cat. #: 1,847 | |
| Chemical compound, drug | Fingolimod | Provided by Bristol-Myers Squibb | N/A | |
| Chemical compound, drug | Gastric Inhibitory Peptide (GIP) | Bachem | Cat. #: H-5645 | |
| Chemical compound, drug | Ghrelin | Tocris | Cat. #: 1,463 | |
| Chemical compound, drug | Glucagon (*Aittaleb et al., 2010*; *Aittaleb et al., 2011*; *Armando et al., 2014*; *Atwood et al., 2011*; *Avet et al., 2020*; *Azzi et al., 2003*; *Bowes et al., 2012*; *Brabet et al., 1998*; *Breton et al., 2010*; *Bünemann et al., 2003*; *Carr et al., 2014*; *Casey et al., 1990*; *Chandan et al., 2021*; *De Haan and Hirst, 2004*; *Devost et al., 2017*; *Fukuhara et al., 2001*; *Galandrin et al., 2007*; *Galés et al., 2005*; *Galés et al., 2006*; *Goupil et al., 2015*; *Hauser et al., 2017*; *Hauser et al., 2022*; *Hoffmann et al., 2005*; *Inoue et al., 2019*; *Jordan et al., 1999*; *Kawamata et al., 2003*; *Kenakin, 2019*; *Kim et al., 2002*) | Bachem | Cat. #: H-6790 | |
| Chemical compound, drug | Glucagon-like peptide-1 GLP-1 (*Bowes et al., 2012*; *Brabet et al., 1998*; *Breton et al., 2010*; *Bünemann et al., 2003*; *Carr et al., 2014*; *Casey et al., 1990*; *Chandan et al., 2021*; *De Haan and Hirst, 2004*; *Devost et al., 2017*; *Fukuhara et al., 2001*; *Galandrin et al., 2007*; *Galés et al., 2005*; *Galés et al., 2006*; *Goupil et al., 2015*; *Hauser et al., 2017*; *Hauser et al., 2022*; *Hoffmann et al., 2005*; *Inoue et al., 2019*; *Jordan et al., 1999*; *Kawamata et al., 2003*; *Kenakin, 2019*; *Kim et al., 2002*; *Kobayashi et al., 2019*; *Laschet et al., 2019*; *Lavoie et al., 2002*; *Leduc et al., 2009*; *Leguay et al., 2021*; *Lu et al., 2014*; *Lutz et al., 2007*) | Bachem | Cat. #: H-6795 | |

*Appendix 1 Continued on next page*

*Appendix 1 Continued*

| Reagent type (species) or resource | Designation | Source or reference | Identifiers | Additional information |
|---|---|---|---|---|
| Chemical compound, drug | Glucagon-like peptide-2 GLP-2 (*Aittaleb et al., 2010*; *Aittaleb et al., 2011*; *Armando et al., 2014*; *Atwood et al., 2011*; *Avet et al., 2020*; *Azzi et al., 2003*; *Bowes et al., 2012*; *Brabet et al., 1998*; *Breton et al., 2010*; *Bünemann et al., 2003*; *Carr et al., 2014*; *Casey et al., 1990*; *Chandan et al., 2021*; *De Haan and Hirst, 2004*; *Devost et al., 2017*; *Fukuhara et al., 2001*; *Galandrin et al., 2007*; *Galés et al., 2005*; *Galés et al., 2006*; *Goupil et al., 2015*; *Hauser et al., 2017*; *Hauser et al., 2022*; *Hoffmann et al., 2005*; *Inoue et al., 2019*; *Jordan et al., 1999*; *Kawamata et al., 2003*; *Kenakin, 2019*; *Kim et al., 2002*; *Kobayashi et al., 2019*; *Laschet et al., 2019*; *Lavoie et al., 2002*; *Leduc et al., 2009*) | Bachem | Cat. #: H-7742 | |
| Chemical compound, drug | Glutamate | Sigma-Aldrich | Cat. #: 49,621 | |
| Chemical compound, drug | GnRH (LH-RH) | Peptides International | Cat. #: PLR-4013 | |
| Chemical compound, drug | Histamine | Tocris | Cat. #: 3,545 | |
| Chemical compound, drug | Kallidin | Anaspec | Cat. #: 22,853(AN) | |
| Chemical compound, drug | Leukotriene B4 (LTB4) | Cayman Chemical | Cat. #: 20,110 | |
| Chemical compound, drug | Leukotriene D4 (LTD4) | Cayman Chemical | Cat. #: 20,310 | |
| Chemical compound, drug | Litocholic acid | Sigma-Aldrich | Cat. #: L6250 | |
| Chemical compound, drug | Melatonin | Bachem | Cat. #: Q-1300 | |
| Chemical compound, drug | MDL 29,951 | Cayman Chemical | Cat. #: 16,266 | |
| Chemical compound, drug | Neuropeptide FF (NPFF) | Tocris | Cat. #: 3,137 | |
| Chemical compound, drug | Neuropeptide Y (NPY) | Bachem | Cat. #: H-6375 | |
| Chemical compound, drug | Nicotinic acid | Abcam | Cat. #: ab120145 | |
| Chemical compound, drug | Nociceptin | Tocris | Cat. #: 910 | |
| Chemical compound, drug | Noradrenaline | Tocris | Cat. #: 5,169 | |
| Chemical compound, drug | Oleoyl-Lysophosphatidic acid (O-LPA) | Sigma-Aldrich | Cat. #: L7260 | |
| Chemical compound, drug | Orexin-A | Bachem | Cat. #: H-4172 | |

*Appendix 1 Continued on next page*

*Appendix 1 Continued*

| Reagent type (species) or resource | Designation | Source or reference | Identifiers | Additional information |
|---|---|---|---|---|
| Chemical compound, drug | Oxytocin | Tocris | Cat. #: 1910 | |
| Chemical compound, drug | Parathyroid Hormone (*Aittaleb et al., 2010*; *Aittaleb et al., 2011*; *Armando et al., 2014*; *Atwood et al., 2011*; *Avet et al., 2020*; *Azzi et al., 2003*; *Bowes et al., 2012*; *Brabet et al., 1998*; *Breton et al., 2010*; *Bünemann et al., 2003*; *Carr et al., 2014*; *Casey et al., 1990*; *Chandan et al., 2021*; *De Haan and Hirst, 2004*; *Devost et al., 2017*; *Fukuhara et al., 2001*; *Galandrin et al., 2007*; *Galés et al., 2005*; *Galés et al., 2006*; *Goupil et al., 2015*; *Hauser et al., 2017*; *Hauser et al., 2022*; *Hoffmann et al., 2005*; *Inoue et al., 2019*; *Jordan et al., 1999*; *Kawamata et al., 2003*; *Kenakin, 2019*; *Kim et al., 2002*; *Kobayashi et al., 2019*; *Laschet et al., 2019*; *Lavoie et al., 2002*; *Leduc et al., 2009*; *Leguay et al., 2021*) | Sigma-Aldrich | Cat. #: P3796 | |
| Chemical compound, drug | Pertussis toxin (PTX) from *Bordetella pertussis* | List Biological Laboratories | Cat. #: 179 A(LB) | |
| Chemical compound, drug | pH (proton) (Hydrochloric acid) | Sigma-Aldrich | Cat. #: 320,331 | |
| Chemical compound, drug | Probenecid | Sigma-Aldrich | Cat. #: P8761 | |
| Chemical compound, drug | Prolume Purple (methoxy e-Coelenterazine; Me-O-e-CTZ) | Nanolight | Cat. #: 369 | |
| Chemical compound, drug | Propionate (sodium salt) | Sigma-Aldrich | Cat. #: P1880 | |
| Chemical compound, drug | Prostaglandin D2 (PGD2) | Cayman Chemical | Cat. #: 12,010 | |
| Chemical compound, drug | Prostaglandin E2 (PGE2) | Sigma-Aldrich | Cat. #: P0409 | |
| Chemical compound, drug | RFamide-related peptide 3 (RFRP3) | Tocris | Cat. #: 4,683 | |
| Chemical compound, drug | Rimonabant | Cayman Chemical | Cat. #: 9000484 | |
| Chemical compound, drug | Saralasin | ApexBio | Cat. #: B5063 | |
| Chemical compound, drug | Serotonin | Cayman Chemical | Cat. #: 14,332 | |
| Chemical compound, drug | SLIGKV-NH2 (PAR2 AP) | Tocris | Cat. #: 3,010 | |
| Chemical compound, drug | SNC80 | Sigma-Aldrich | Cat. #: S2812 | |
| Chemical compound, drug | Somatostatin-14 | Bachem | Cat. #: H-6276 | |
| Chemical compound, drug | Sphingosine 1-phosphate | Cayman | Cat. #: 62,570 | |

*Appendix 1 Continued on next page*

*Appendix 1 Continued*

| Reagent type (species) or resource | Designation | Source or reference | Identifiers | Additional information |
|---|---|---|---|---|
| Chemical compound, drug | TFLLR-NH2 (PAR1 AP) | Tocris | Cat. #: 1,464 | |
| Chemical compound, drug | TRV027 | Provided by Bristol-Myers Squibb | N/A | |
| Chemical compound, drug | UBO-QIC (FR900359) | Institute for Pharmaceutical Biology of the University of Bonn | N/A | |
| Chemical compound, drug | Undecanoic acid | Sigma-Aldrich | Cat. #: 171,476 | |
| Chemical compound, drug | Urocortin II | Phoenix Pharmaceutical | Cat. #: 019–30 | |
| Chemical compound, drug | UTP | Sigma-Aldrich | Cat. #: U1006 | |
| Chemical compound, drug | Vasoactive Intestinal Peptide (VIP) | Tocris | Cat. #: 1911 | |
| Chemical compound, drug | WB4101 | Tocris | Cat. #: 946 | |
| Chemical compound, drug | WIN55,212–2 | Enzo Life Sciences | Cat. #: BMLCR105 | |
| Chemical compound, drug | YM-254890 | Wako Pure Chemical Industries (Fujifilm) | Cat. #: 257–00631 | |
| Chemical compound, drug | Zinc chloride (Zn2+) | Sigma-Aldrich | Cat. #: 229,997 | |
| Software, algorithm | Prism, Version 8.3 | GraphPad | | |
| Software, algorithm | MATLAB, Version R2019b | MathWorks | | |
| Software, algorithm | ImageJ, Version 1.52 a | NIH https://imagej.nih.gov/ij/ | | |
| Software, algorithm | Scipy, Version 1.4.1 | https://www.scipy.org | | |
| Other | 96 W white plate | Greiner Bio-one | Cat. #: 655,083 | |
| Other | 96 W black plate, clear-bottom | Greiner Bio-one | Cat. #: 655,090 | |
| Other | OptiPlate-384, White Opaque 384-well Microplate | Perkin Elmer | Cat. #: 6007290 | |
| Other | 35 mm poly-d-lysine-coated glass-bottom culture dishes | Mattek | Cat. #: P35GC-1.5–14 C | |
| Other | Microplate washer | BioTek Instruments | Cat. #: 405TSUS | |
| Other | D300e Digital Dispenser | Tecan | | |
| Other | T8 + Dispensehead Cassettes | Hp (Tecan) | Cat. #: 30097370 | |
| Other | Synergy NEO Luminescence microplate reader | BioTek Instruments | | |
| Other | FlexStation 2 Multi-mode, auto-pipetting microplate reader | Molecular Devices | | |
| Other | Inverted microscope | Nikon Eclipse Ti-U | | |
| Other | x60 objective lens | Nikon CFI Apochromat TIRF | | |
| Other | EMCCD camera | Nuvu HNu 512 | | |

