## [Editor Report]

The authors describe a novel set of biosensors to assess the coupling specificity of 100 therapeutically relevant G proteins-coupled receptors (GPCRs) to various G proteins. The utility of the assay system is well-supported by the data. These tools are likely to be useful for many specific studies of individual receptors, including efforts to discover ligands that display functional selectivity (bias) between G protein pathways or between G proteins and arrestins. The work provides a rich repository of data informing on the possible effector coupling of 100 GPCRs and a set of analytical tools that could guide the development of new drugs, including efforts to discover ligands that display functional selectivity (bias) between G protein pathways or between G proteins and arrestins.

---

## [Decision Letter]

**Decision letter after peer review:**

Thank you for submitting your article "Effector membrane translocation biosensors reveal G protein and βarrestin coupling profiles of 100 therapeutically relevant GPCRs" for consideration by *eLife*. Your article has been reviewed by 3 peer reviewers, and the evaluation has been overseen by a Reviewing Editor and Richard Aldrich as the Senior Editor. The following individual involved in review of your submission has agreed to reveal their identity: Roger Sunahara (Reviewer #1).

Essential revisions:

The reviewers agree that the authors have generated a useful set of reagents and analysis tools for investigating G protein coupling by GPCRs, and is in principle appropriate as a Tools and Resources contribution, but it was felt that the strengths and weaknesses need to be presented in a more cautious manner. The need for overexpression of receptors, G proteins and effectors (points 2, 3 and 4) is not adequately discussed, and the authors need to make clear to the general reader that the assays indicate what GPCR-G couplings are possible, as distinct from what happens physiologically. It is also felt that the "non modified" G protein argument is overstated given that Gs is modified; nonetheless, there is very likely significant advantage in using unmodified receptors.

1. The authors should show an example in which this assay system shows a clear benefit versus other available high-throughput assays. The authors refer to the companion paper that presents a detailed analysis of selectivity profiles and comparison with existing data sets. It would be better if this analysis (or at least the main points) was simply included in the present manuscript, as this might better demonstrate cases where the EMTA assays outperform other assays, a key claim of the present manuscript. Moreover, the authors say that their "comparative analysis" (line 275) revealed a number of couplings not found in IUPHAR or Inoue, but simple inspection of the data does the same thing. For completeness, that should at least mention that their dataset also lacks some couplings found in other datasets.

2. The authors discuss the caveats with the EMTA approach, stating that the assays require overexpression of both G protein isoforms and receptors, although they can be used with endogenous receptors. It is unclear how overexpression of GPCRs and G proteins could influence the engagement of downstream effectors and possible mislead interpretation about potential biased signaling or polypharmacology. The authors need to present a balanced discussion of the relationship between receptor and G proteins overexpression. Notably, the authors have previously claimed that ligand biased signaling is dependent on cellular context, where various levels of expression of receptor, G proteins, effectors are found. With this in mind, how valuable is the information presented since the measurements were acquired in a non-natural context? Some experimental examples varying receptor and G protein expression and monitoring effector engagement would be ideal to address these issues.

3. Related to point 2, the assays are limited to a subset of effectors. The authors still need to address whether overexpression of the effector-sensor also biases the response. Typical endogenous effectors are expressed at very low levels owing to their roles as catalysts. In this assay, however, overexpression of the sensor is required as a BRET pair. Thus, are the authors confident that over expression of the effector-sensor does not bias the coupling response, either by altering the potency or efficacy of agonist activation?

4. The authors state that effector coupling between G proteins from the same subclass are observed (i.e. different Galphai, G12 vs G13 etc…)? In order to claim this, the authors would need to carefully monitor the level of expression of the G proteins (if not possible by WB then qPCR). For example, in figure 2C, the authors present data that CB1 seems to couple to Galpha 13 more than Galpha 12, two G proteins that have been thought to be functionally equivalent. Could G protein isoforms have different affinities for the biosensors?

5. The authors are for the most part clear about the fact that Gs needed to be modified in their experiments, but in several places they gloss over this in a way that tends to overstate their results. For example, on line 219 they refer to theirs as the first dataset to use "unmodified GPCRs and G proteins" in the same sentence as promiscuous coupling to 2,3 or 4 families. Some of this promiscuous coupling must have been demonstrated with modified Gs. In general the authors should be cautious about the broad claim that the assay uses unmodified G proteins when one of the four major families is indeed modified.

6. The authors considered the effects of glutamate in media for the mGluR studies, but the authors use A1 adenosine receptors to demonstrate constitutive activity, with larger inverse agonist responses than agonist responses. However, a limitation of this example is the ubiquity of adenosine in cell cultures; studying adenosine receptors in HEK cells is not straightforward without testing the effects of adenosine deaminase to metabolize adenine-containing molecules released into the media by the cells themselves. Either deaminase treatment should be performed for this experiment, or a better receptor example with known constitutive activity (e.g. beta2AR or CB1), where the possibility of an endogenous ligand being present is more remote.

7. When discussing the coupling efficiency of GPCRs to subsets of Galpha subunits, the authors highlight the adenosine A2A receptor as one which preferentially couples to Galpha15. The data in the table really suggest that the A2B receptor might be more selective for Galpha15 then A2A.

8. To compare the receptor coupling efficiencies, the authors normalized the dose-response curves of the agonists for each G protein (or arrestin) against a reference receptor on that given pathway. Similarly, the authors illustrated the Emax for each coupling relationship as a normalized response to a reference receptor that yields a maximal response. These are rational interpretations for comparison purposes. For physiological and pharmacological purposes, it might be useful if the authors expressed the coupling specificity in a slightly different manner. It would be useful for readers to see how an effector response would be for a non-canonical coupling event relative to a cognate coupling relationship. For example, what would be the effector response for norepinephrine stimulation of beta1AR with the non-cognate G proteins using an EC50 concentration determined from the Galphas (cognate) measurements? Here the physiological concentrations of norepinephrine that are necessary for Galphas activation (coupling) can be compared to the coupling to other Galphas (or arrestin) at the same concentration. At least under non-pathological conditions, coupling with hormone or even therapeutically relevant drug concentrations could then be compared fairly. The data are already collected by the authors and assembling a table illustrating this comparison should not be too difficult. This sort of interpretation/presentation should be quite informative.

9. The normalization procedure that compares all responses as a percentage of baseline may have unintended consequences for receptors with high constitutive activity or where endogenous ligands may be present in HEK cell cultures (e.g. adenosine, 5-HT, glutamate). Basal responses may be high for the most efficient G proteins but lower for secondary couplers, which could paradoxically give the latter a larger response window. This might skew the results, e.g. does A1 really activate G12 better than Gi subunits, or rather does the G12 sensor simply have a higher window due to lower baseline. This might also explain why Gi looks like a better coupler to mGluR5 than Gq; perhaps the Gq signal is already close to saturation, as glutamate is also difficult to remove.

10. Line 418 the authors claim that EMTA does not require amplification. It's unclear precisely what they mean in this context, but amplification could certainly occur with this assay system, as a single active receptor could maintain many G proteins in the active state. This might be especially true if the EMTA probes themselves prevent binding of RGS proteins, thus prolonging the active lifetime of each G protein. It is now clear that assays that lack amplification are advantageous for studies of ligand bias, therefore the authors should be careful to avoid making this claim.

11. Line 505 the authors refer to the use of EMTA for HTS. BRET is rarely used for true HTS, so this statement might be softened somewhat.

12. The authors should directly address a recent study showing that PDZ-RhoGEF is activated by Gi subunits (https://www.biorxiv.org/content/10.1101/2021.07.15.452545v1.full). Were all experiments with their PDZ-RhoGEF sensor done in the presence of PTX, as in Figure 2C?

13. The authors claim that EMTA allows for imaging of the spatiotemporal activation of the different Biosensors. Although this is convincing for video 2 and video 3, it is less so for video 1 (galphaI). Can the authors provide additional information or an alternative representative video?

14. On lines 300-301, the authors mention that HEK293 cells do not express Galpha15 and that the EP2 and opioid receptors do not couple to Gq/11 family members but couple to Galpha15. However, in Figure 3 supp 3B, there is a clear calcium response upon PGE2 or Dynorphin exposure. Galpha15 expression only potentiated the response. How is calcium signaling activated if it is not Galphaq/11 mediated?

15. The authors suggest that the cross-activation of the D2 and alpha2 receptors by noradrenaline and serotonin is direct (through direct engagement of D2 and alpha2) as the response is inhibited by D2 and alpha2 antagonist. This conclusion would be further strengthened if the response was not inhibited by adrenergic or serotoninergic antagonists (as done with the muscarininc antagonist for CB1-CB2 cross talk).

*Reviewer #1 (Recommendations for the authors):*

There are some issues that should to addressed by the authors. None are deal breakers but they should be addressed, most likely possible without any further experimentation.

Comments:

The authors nicely state the caveats with the EMTA approach stating that the assays require overexpression of both G protein isoforms and receptors, although can be used with endogenous receptors. As well, they state that the assays are limited to a subset of effectors. The authors still need to address whether overexpression of the effector-sensor also biases the response. Typical endogenous effectors are expressed at very low levels owing to their roles as catalysts. In this assay, however, overexpression of the sensor is required as a BRET pair. Thus, are the authors confident that over expression of the effector-sensor does not bias the coupling response, either by altering the potency or efficacy of agonist activation?

The coupling results are interesting with the adenosine receptors. It appears that most of the adenosine receptors tested displayed high constitutive activity. Is this due to adenosine released in the media? Studying adenosine receptors in HEK cells is not straight forward without testing the effects of adenosine deaminase to metabolize adenine-containing molecules released into the media by the cells themselves. The authors did consider the effects of glutamate in media for the mGluR studies but I'm not certain they considered it for the adenosine receptors.

Also, when discussing the coupling efficiency of GPCRs to subsets of Galpha subunits, the authors highlight the adenosine A2A receptor as one which preferentially couples to Galpha15. The data in the table really suggest that the A2B receptor might be more selective for Galpha15 then A2A.

To compare the receptor coupling efficiencies the authors normalized the dose-response curves of the agonists for each G protein (or arrestin) against a reference receptor on that given pathway. Similarly, the authors illustrated the Emax for each coupling relationship as a normalized response to a reference receptor that yields a maximal response. These are rational interpretations for comparison purposes. For physiological and pharmacological purposes it might be useful if the authors expressed the coupling specificity in a slightly different manner. It would be useful for readers to see how an effector response would be for a non-canonical coupling event relative to a cognate coupling relationship. For example, what would be the effector response for norepinephrine stimulation of beta1AR with the non-cognate G proteins using an EC50 concentration determined from the Galphas (cognate) measurements? Here the physiological concentrations of norepinephrine that are necessary for Galphas activation (coupling) can be compared to the coupling to other Galphas (or arrestin) at the same concentration. At least under non-pathological conditions, coupling with hormone or even therapeutically relevant drug concentrations could then be compared fairly. The data are already collected by the authors and assembling a table illustrating this comparison should not be too difficult. This sort of interpretation/presentation should be quite informative.

*Reviewer #2 (Recommendations for the authors):*

The authors are for the most part clear about the fact that Gs needed to be modified in their experiments, but in several places they gloss over this in a way that tends to overstate their results. For example, on line 219 they refer to theirs as the first dataset to use "unmodified GPCRs and G proteins" in the same sentence as promiscuous coupling to 2,3 or 4 families. Some of this promiscuous coupling must have been demonstrated with modified Gs.

The authors refer to a companion paper that presents a detailed analysis of selectivity profiles and comparison with existing data sets. It would be better if this analysis (or at least the main points) was simply included in the present manuscript, as this might better demonstrate cases where the EMTA assays outperform other assays, a key claim of the present manuscript.

The authors say that their "comparative analysis" (line 275) revealed a number of couplings not found in IUPHAR or Inoue, but simple inspection of the data does the same thing. For completeness, that should at least mention that their dataset also lacks some couplings found in other datasets.

The term "therapeutically-relevant" is not defined, isn't necessary, and seems a stretch for many of the receptors sampled here, many of which are not demonstrated therapeutic targets.

The authors make frequent use of the term "engage". This is semantics, but the term is not as precise as "bind" or "activate” and seems closer to "bind". In this case I much prefer "activate", after all a key advantage of the EMTA format is that G proteins must be activated, not just bound or engaged by a GPCR.

The authors use A1 adenosine receptors to demonstrate constitutive activity, with larger inverse agonist responses than agonist responses. However, a limitation of this example is the ubiquity of adenosine in cell cultures, which often requires adenosine deaminase treatment to remove completely. A better example might be another receptor with known constitutive activity (e.g. beta2AR or CB1), where the possibility of an endogenous ligand being present is more remote.

Line 364 refers to prior observation of pleiotropic activation of monoamine receptors. A citation is needed.

Line 418 the authors claim that EMTA does not require amplification. It's unclear precisely what they mean in this context, but amplification could certainly occur with this assay system, as a single active receptor could maintain many G proteins in the active state. This might be especially true if the EMTA probes themselves prevent binding of RGS proteins, thus prolonging the active lifetime of each G protein. It is now clear that assays that lack amplification are advantageous for studies of ligand bias, therefore the authors should be careful to avoid making this claim.

Line 505 the authors refer to the use of EMTA for HTS. BRET is rarely used for true HTS, so this statement might be softened somewhat.

The authors should directly address a recent study showing that PDZ-RhoGEF is activated by Gi subunits (https://www.biorxiv.org/content/10.1101/2021.07.15.452545v1.full). Were all experiments with their PDZ-RhoGEF sensor done in the presence of PTX, as in Figure 2C?

The normalization procedure which compares all responses as a percentage of baseline may have unintended consequences for receptors with high constitutive activity or where endogenous ligands may be present in HEK cell cultures (e.g. adenosine, 5-HT, glutamate). Basal responses may be high for the most efficient G proteins but lower for secondary couplers, which could paradoxically give the latter a larger response window. This might skew the results, e.g. does A1 really activate G12 better than Gi subunits, or rather does the G12 sensor simply have a higher window due to lower baseline. This might also explain why Gi looks like a better coupler to mGluR5 than Gq; perhaps the Gq signal is already close to saturation, as glutamate is also difficult to remove.

*Reviewer #3 (Recommendations for the authors):*

1) The authors highlight that one of the strengths of EMTA is the use of untagged receptor and G proteins but fail to present an example where this constitutes a true benefit when compared to other biosensors or signaling assays.

2) although EMTA functions in some endogenous cases where receptor expression and effector coupling are optimal, in most contexts overexpression of both GPCRs and G proteins is required. It is unclear how overexpression of GPCRs and G proteins could influence the engagement of downstream effectors and possible mislead interpretation about potential biased signaling or polypharmacology. The authors need to present a careful analysis of the relationship between receptor and G proteins overexpression (using qPCR) and effector engagement? This could be done by varying receptor and g protein expression and monitoring effector engagement.

3) The authors have previously claimed that ligand biased signaling is dependent on cellular contexts, where various levels of expression of receptor, G proteins, effectors are found. With this in mind, how valuable is the information presented since the measurements were acquired in a non-natural context?

4) In figure 2B on the right, it appears that PTX leads to a significant inhibition of the G protein coupling (non Galphai) in response to GnRH, is this significant? The authors state on lines 128-129 that the response is not affected by PTX. This should be clarified.

5) The authors state that effector coupling between G proteins from the same subclass are observed (i.e. different Galphai, G12 vs G13 etc…)? In order to claim this, the authors need to carefully monitor the level of expression of the G proteins (if not possible by WB then qPCR). For example, in figure 2C, the authors present data that CB1 seems to couple to Galpha 13 more than Galpha 12, two G proteins that have been thought to be functionally equivalent. Could G protein isoforms have different affinities for the biosensors?

6) The authors claim that EMTA allows for imaging of the spatiotemporal activation of the different Biosensors. Although this is convincing for video 2 and video 3, it is less so for video 1 (galphaI). Can the authors provide additional information or alternative representative video?

7) On lines 300-301, the authors mention that HEK293 cells do not express Galpha15 and that the EP2 and opioid receptors do not couple to Gq/11 family members but couple to Galpha15. However, in Figure 3 supp 3B, there is a clear calcium response upon PGE2 or Dynorphin exposure. Galpha15 expression only potentiated the response. How is calcium signaling activated if it is not Galphaq/11 mediated?

8) The authors suggest that the cross-activation of the D2 and alpha2 receptors by noradrenaline and serotonin is direct (through direct engagement of D2 and alpha2) as the response is inhibited by D2 and alpha2 antagonist. This conclusion would be further strengthened if the response was not inhibited by adrenergic or serotoninergic antagonists (as done with the muscarininc antagonist for CB1-CB2 cross talk).

9) Perhaps the most interesting new biology uncovered in this study is the indirect activation of CB1 and CB2 signaling by muscarinic agonist, a mechanism the authors suggest could involve the secretion of a cannabinoid ligand by muscarinic receptor activation. The authors however stop short of directly demonstrating that this is the case.

---

## [Author Response]

Essential revisions:The reviewers agree that the authors have generated a useful set of reagents and analysis tools for investigating G protein coupling by GPCRs, and is in principle appropriate as a Tools and Resources contribution, but it was felt that the strengths and weaknesses need to be presented in a more cautious manner. The need for overexpression of receptors, G proteins and effectors (points 2, 3 and 4) is not adequately discussed, and the authors need to make clear to the general reader that the assays indicate what GPCR-G couplings are possible, as distinct from what happens physiologically. It is also felt that the "non modified" G protein argument is overstated given that Gs is modified; nonetheless, there is very likely significant advantage in using unmodified receptors.

We thank the reviewers for their positive comments concerning the usefulness of the reagent set presented and validated in our study. We agree with the reviewers that, as for most assays, the EMTA platform has its limitations. Concerning, the notion that the assays indicate what GPCR-G couplings are possible, and not necessarily what happens physiologically, we agree with the reviewers and we had mentioned this point in the discussion of the original manuscript in the following way:

‘A second potential caveat of EMTA is the use of overexpressed GPCRs and G proteins. Some of the responses detected could indeed result from favorable stoichiometries that may not exist under physiological conditions. It follows that the profiling represents the coupling possibilities of a given GPCR and not necessarily the coupling that will be observed in all cell types.’ (lines 472-476 of the revised manuscript).

To emphasize it even further, we now also addressed this point in a more general way as early as the introduction (lines 82-86 of the revised manuscript) and more directly linked it to the EMTA assays in the Results section (line 195 of the revised manuscript).

The reviewer is right that the possibility to use the assay with endogenous levels of receptors or G proteins (discussed below) should not detract from the fact that, in our manuscript, the EMTA platform was mainly used with heterologously expressed receptors and G proteins, since the goal was to perform a large-scale scanning of the G protein coupling potentials for a large collection of GPCRs. Any couplings observed in such high-throughput studies will require further validation to conclude on their physiological relevance in cells or tissues of interest. This has now been reiterated in the discussion on lines 476-478. Yet, it should be noted that the assays for G_q/11_ and, G_i/o_ families can be used in the absence of heterologously expressed G proteins (lines 127-131 of the revised manuscript). However, as we have now indicated on lines 131-133, it is impossible under these conditions to distinguish which members of the family is activated by the receptor. As also noted on lines 442-454 of the revised manuscript, the assay can be used to profile endogenously expressed receptor and G proteins, with no need for overexpression of either the receptor or G proteins, given a sufficient endogenous expression level.

Concerning the special case of G_s_ that needs to be modified to detect its activation in an EMTA format, this had been mentioned in the original discussion:

‘Another limitation is the lack of a soluble effector protein selective for activated Gαs thus requiring tagging of the Gα_s_ subunit (Figure 1B, bottom) and monitoring its dissociation from the plasma membrane. Yet, our data show that this translocation reflects Gs activation state, justifying its use in a G protein activation detection platform’ (lines 503-507).

For clarity sake, we have now stated the exception of G_s_ everytime we evoked the ability of detecting G protein activation without modification of either Gα or Gβγ proteins.

1. The authors should show an example in which this assay system shows a clear benefit versus other available high-throughput assays. The authors refer to the companion paper that presents a detailed analysis of selectivity profiles and comparison with existing data sets. It would be better if this analysis (or at least the main points) was simply included in the present manuscript, as this might better demonstrate cases where the EMTA assays outperform other assays, a key claim of the present manuscript. Moreover, the authors say that their "comparative analysis" (line 275) revealed a number of couplings not found in IUPHAR or Inoue, but simple inspection of the data does the same thing. For completeness, that should at least mention that their dataset also lacks some couplings found in other datasets.

We would like to respectfully emphasize that we did not meant to say that our assay system universally outperforms others. To more clearly present the advantages and limitations of EMTA in a more balance matter, the section of advantages, limitations and caveats have been significantly expanded (lines 82-86, 127-133, 163-167, 173-177, 195, 343-347, 442-454, 476-478, 483-501, 589-592 and 597-600 of the revised manuscript).

Yet, to directly address the question of the reviewer regarding examples in which EMTA shows a clear benefit, new data presenting side-by-side comparison of the signals detected with EMTA *vs.* BRET assays based on Gαβγ dissociation (Gαβγ) (Gales et al., Nat Methods 2005; Gales et al., Nat Struct Mol Biol 2006; Olsen et al., Nat Chem Biol 2020) had been added. As shown in new Figure 2—figure supplement 5, EMTA generated significantly larger assay windows than Gαβγ assays for the 6 Gα subunits tested for the 8 receptors selected (D_2_, GIP, PTH1, M_3_, ET_A_, B_1_, FP or Cys-LT_2_). In some cases, a robust response could be observed only with EMTA. These data are now presented in the Results section (lines 173-177).

Given that some of the G_12/13_ couplings had not been reported by Inoue et al., or in the GtP dataset, the validity of these couplings was confirmed using orthogonal assays downstream of G_12/13_ in the absence of heterologously expressed G proteins. As shown in modified Figure 3—figure supplement 4A, G_12/13_ activation by FP and Cys-LT_2_ was detected using the PKN-based (Namkung et al., Sci Signal 2018) and MyrPB-Ezrin-based (Leguay et al., J. Cell Sci 2021) BRET biosensors, two biosensors detecting Rho activation downstream of G_12/13_; a response not affected by G_q/11_ inhibition. These data are now described in the revised manuscript (lines 314-316). Such confirmation of new couplings compared to Inoue *et al.,* and GtP dataset was also provided for G_15_ by monitoring calcium-induced responses (see Figure 3—figure supplement 4B).

As requested by the reviewer, although an extensive comparison between datasets is presented in the companion paper, we included a new table directly comparing couplings identified in our study *vs.* that of Inoue *et al.,* (Supplementary File 4A). This table clearly shows that EMTA not only detected couplings that were not reported in Inoue *et al.*, but also did not detect some of the couplings seen in Inoue *et al.* This is now discussed more extensively on lines 288-292. Direct comparison with the GtP dataset is less straightforward since all members of a given family are clustered in one category in this database. Yet, when considering the family clusters, some differences can be observed between the couplings identified in our study, and those reported in GtP database. This is now also discussed in lines 292-296 and in Supplementary File 4B, and a more detailed analysis can be found in the companion paper (Hauser *et al.*, 2021 bioRxiv. doi: https://doi.org/10.1101/2021.09.07.459250).

2. The authors discuss the caveats with the EMTA approach, stating that the assays require overexpression of both G protein isoforms and receptors, although they can be used with endogenous receptors. It is unclear how overexpression of GPCRs and G proteins could influence the engagement of downstream effectors and possible mislead interpretation about potential biased signaling or polypharmacology. The authors need to present a balanced discussion of the relationship between receptor and G proteins overexpression. Notably, the authors have previously claimed that ligand biased signaling is dependent on cellular context, where various levels of expression of receptor, G proteins, effectors are found. With this in mind, how valuable is the information presented since the measurements were acquired in a non-natural context? Some experimental examples varying receptor and G protein expression and monitoring effector engagement would be ideal to address these issues.

Concerning the idea that the signaling profile of a receptor can be influence by the expression levels of its downstream transducers and effectors (a concept known as system bias), the reviewer is right. Again, the point of the manuscript is not to determine the signaling profiles of a given receptor in specific physiologically relevant tissues but rather to provide a broad view of the coupling potentials of a large number of receptors. As indicated in the response to the previous comments, this potential caveat of the assays has been emphasized in lines 82-86, 127-133, 465-470, 472-501, 589-592 and 597-600 of the revised manuscript. Given that for all receptors tested the same amount of a given G protein was used to assess their coupling potential, the relative propensity of each of these receptors to activate the different G proteins can be assessed. Such comparison is greatly facilitated by the double normalization used in the study (see lines 208-216) since it allows ranking the coupling propensity of the receptors first as a function of the receptor which shows the strongest coupling to a specific G protein subtype, and then establishing the maximal response observed for a given G protein subtype as the reference for all G proteins activated by a given receptor. This should greatly diminish, if not obliterate, the influence of the G protein and receptor expression levels. Nevertheless, we added a new supplemental figure (Figure 2—figure supplement 3A), demonstrating that varying G protein expression level can modify the amplitude of the response (as expected since the purpose of our platform is to identifying specific Gα subunits activated by a given receptor by overexpressing them) but not the potency (pEC_50_) of ligand-promoted effectors recruitment by activated-G proteins (Supplementary File 1B). Similarly, varying GPCR expression level only modified the amplitude of the response but not the pEC_50_ (Figure 2—figure supplement 3B and Supplementary File 1C). These results are now discussed in the revised manuscript (lines 163-167 and 487-489). It should also be noted that in the original manuscript (lines 159-163 of the revised manuscript), we provided evidence that no competition with endogenous G proteins occurs to a significant extent in the heterologous expression configuration, since the potencies of the responses to a given G protein subtype were not affected by the genetic deletion of the different G protein family members (see lines 490-492 of the revised manuscript, Figure 2—figure supplement 1 and Supplementary File 1A).

The question of ligand-biased signaling is a different one. This is a comparison between the propensity of two ligands to lead to the activation of two different pathways. This is theoretically an intrinsic property of the ligand-receptor pair and should be independent of the absolute expression of the partners as long as the two ligands are compared under equivalent conditions. No systematic analysis of ligand-bias has been attempted in the present study. The only reference to ligand-biased signaling concerns an example that was given for type-1 angiotensin II receptor (AT_1;_ see Figure 5B and text lines 349-353 of the manuscript) to illustrate that biases previously reported could be detected using EMTA. We also provided an example showing that mutations of a given receptor (i.e.: GPR17) could affect the signaling preference of this receptor (See Figure 5C and text lines 353-359 of the manuscript). Again, these experiments were performed at equal quantities of effectors and cannot be explain by difference in receptor expression levels since signaling to specific pathway was either untouched or compromised in a mutation-depend manner.

3. Related to point 2, the assays are limited to a subset of effectors. The authors still need to address whether overexpression of the effector-sensor also biases the response. Typical endogenous effectors are expressed at very low levels owing to their roles as catalysts. In this assay, however, overexpression of the sensor is required as a BRET pair. Thus, are the authors confident that over expression of the effector-sensor does not bias the coupling response, either by altering the potency or efficacy of agonist activation?

The reviewer is right in stating that the RlucII-fused effector constructs need to be heterologously expressed in the EMTA assay as they are part of the biosensors. It should however be specified that only part of the PDZ and p63 sequences are included in the biosensors. For the PDZ biosensor, the sequence fused to RlucII include the G_12/13_ protein binding domain but is lacking the PDZ domain involved in protein-protein interaction, the actin-binding domain and the DH/PH domains involved in GEF activity and RhoA activation (Mohamed Aittaleb et al., Molecular Pharmacology 2010). For the p63 biosensor, the sequence fused to RlucII included the G_q_ binding domain (which is included in the PH domain), but lacked the N-term part, containing the palmitoylation sites maintaining p63 to plasma membrane, and part of the DH domain involved in RhoA binding/activation (Mohamed Aittaleb et al., Molecular Pharmacology 2010; Mohamed Aittaleb et al., JBC 2011). For Rap1GAP, a bigger part of the protein was kept as two regions are important for the interaction with G_i/o_ proteins. However, mutations of protein kinase A phosphorylation sites (i.e.: S437A/S439A/S441A) were introduced in order to eliminate any G_s_-mediated Rap1GAP recruitment to the plasma-membrane (McAvoy et al., PNAS 2009). Altogether, the use of truncated part and/or modified version of these effector limits the possibilities of interference with other components of the signaling machinery, and serves essentially as a binding detector of the active forms of the G proteins. These precisions have now been added to the Material and Method section of the revised manuscript (see lines 589-592 and 597-600).

Although the ratio of effector-RlucII to rGFP-CAAX is important to define the assay window (maximal response), as is the case in any BRET assay, the expression level of the RlucII-effector should not affect the measured pEC_50_ of the response. In agreement with this statement, similar pEC_50_ were obtained for different levels of effector-RlucII expression as illustrated in the new Figure 2—figure supplement 3C, with corresponding pEC_50_ reported in Supplementary File 1D, and discussed in the section validating the assay platform (lines 163-167 and 499-501 of revised manuscript). These data indicate that heterologous expression of the biosensors did not influence the propensity of a receptor to activate a given G protein. Consistent with this notion, recruitment of the effectors was entirely G protein selective (Figure 2—figure supplement 1 and Figure 2—figure supplement 2) in agreement with the documented selectivity of the G protein effectors used.

4. The authors state that effector coupling between G proteins from the same subclass are observed (i.e. different Galphai, G12 vs G13 etc…) ? In order to claim this, the authors would need to carefully monitor the level of expression of the G proteins (if not possible by WB then qPCR). For example, in figure 2C, the authors present data that CB1 seems to couple to Galpha 13 more than Galpha 12, two G proteins that have been thought to be functionally equivalent. Could G protein isoforms have different affinities for the biosensors?

The reviewer is right that the expression levels of the G proteins belonging to the same subclass (ex: G_12_ and G_13_) could theoretically lead to apparent difference in coupling that would be driven by mass action. However, it should be noted that all experiments were carried out using the same amount of G protein cDNA (using the same plasmid backbone) for all receptors tested. Yet, differences could still occur. This is why we used the double normalization approach (see lines 208-216 of the revised manuscript) that allows ranking the coupling propensity of the receptors as a function of the receptor which shows the strongest coupling to a specific G protein subtype. This should greatly diminish, if not obliterate, the influence of the G protein expression level.

In the specific case of G_12_ and G_13_, the fact that we observe receptors that couple to G_13_ but not G_12_ (14) and reciprocally to G_12_ but not G_13_ (6) indicates that a simple difference in expression levels cannot explain the selectivity between the subtypes. Similarly, for receptors that were found to couple to both G_12_ and G_13_, some showed greater potencies for G_12_ whereas others for G_13_.

It should also be noted that western blot analyses of the G protein expression levels were presented in the original Figure 2—figure supplement 4 (Figure 2—figure supplement 6 of the revised manuscript) to illustrate the level of overexpression *vs.* endogenous levels for each of the G protein subtype. However, this cannot be use to compare the expression levels of the different G proteins since the different antibodies may have different affinities/avidities. qPCR would also be an inadequate estimate as it would not consider differences in the translation and stability of the individual G proteins.

As indicated above, we believed that our double normalization methods alleviate the possible distortions promoted by different expression levels. The lack of effect of increasing G protein levels on the pEC_50_ of ligand-promoted G protein activation observed in the new Figure 2—figure supplement 3A, also suggests that the differences observed do not result from different expression levels (see response to question 2).

Another variable that could impact the apparent preference among G protein of the same subclass that is mitigated by the double normalization, is the possibility that the effectors-RlucII constructs may have different affinities for the members of the G protein family that they recognize. This was already acknowledged in the original discussion:

‘A potential caveat of EMTA is the use of common downstream effectors for all members of a given G protein family. Indeed, one cannot exclude the distinct members of a given family may display different relative affinities for their common effector. However, such differences are compensated by our data normalization that establishes the maximal response observed for a given subtype as the reference for this pathway (Figure 3A).’ (lines 465-470 of the revised manuscript).

5. The authors are for the most part clear about the fact that Gs needed to be modified in their experiments, but in several places they gloss over this in a way that tends to overstate their results. For example, on line 219 they refer to theirs as the first dataset to use "unmodified GPCRs and G proteins" in the same sentence as promiscuous coupling to 2,3 or 4 families. Some of this promiscuous coupling must have been demonstrated with modified Gs. In general the authors should be cautious about the broad claim that the assay uses unmodified G proteins when one of the four major families is indeed modified.

We carefully revised the text not to lead the readers to believe that all G protein activities could be detected without modification of the Gα subunit. In every case, we made it clear that this was possible for all Gα subunits except Gα_s_.

6. The authors considered the effects of glutamate in media for the mGluR studies, but the authors use A1 adenosine receptors to demonstrate constitutive activity, with larger inverse agonist responses than agonist responses. However, a limitation of this example is the ubiquity of adenosine in cell cultures; studying adenosine receptors in HEK cells is not straightforward without testing the effects of adenosine deaminase to metabolize adenine-containing molecules released into the media by the cells themselves. Either deaminase treatment should be performed for this experiment, or a better receptor example with known constitutive activity (e.g. beta2AR or CB1), where the possibility of an endogenous ligand being present is more remote.

We agree with the reviewer that the high level of basal activity observed for the A_1_ adenosine receptor could result from activation by adenosine in cell culture medium rather than constitutive activity. Yet, despite the fact that adenosine was found to have the same potency to activate A_1_ and A_3_ receptors (see Figure 5—figure supplement 1A), high basal activity was observed for A_1_ but not A_3_, supporting the notion that it is truly constitutive activity that has been observed. This is now included in the manuscript lines 336-342. Nevertheless, we chose to also provide another example where inverse efficacy could be observed with the EMTA assay. A new panel in Figure 5A indeed describes the constitutive activity of CB_1_ on G_z_ activation and the inverse agonist activity of rimonabant, and is described lines 343-347.

7. When discussing the coupling efficiency of GPCRs to subsets of Galpha subunits, the authors highlight the adenosine A2A receptor as one which preferentially couples to Galpha15. The data in the table really suggest that the A2B receptor might be more selective for Galpha15 then A2A.

When we referred to the selectivity of A_2A_ in the manuscript (line 370 of the revised manuscript), it was done in the context of the G_z_/G_15_ dual biosensor. It was presented to provide an example of a receptor that couples to only one of the two pathways (G_15_ and not G_z_) and which activity could be detected by the dual biosensor. Nevertheless, the reviewer is right that this is also the case for A_2B_, which we now added as an additional example (line 370).

8. To compare the receptor coupling efficiencies, the authors normalized the dose-response curves of the agonists for each G protein (or arrestin) against a reference receptor on that given pathway. Similarly, the authors illustrated the Emax for each coupling relationship as a normalized response to a reference receptor that yields a maximal response. These are rational interpretations for comparison purposes. For physiological and pharmacological purposes, it might be useful if the authors expressed the coupling specificity in a slightly different manner. It would be useful for readers to see how an effector response would be for a non-canonical coupling event relative to a cognate coupling relationship. For example, what would be the effector response for norepinephrine stimulation of beta1AR with the non-cognate G proteins using an EC50 concentration determined from the Galphas (cognate) measurements ? Here the physiological concentrations of norepinephrine that are necessary for Galphas activation (coupling) can be compared to the coupling to other Galphas (or arrestin) at the same concentration. At least under non-pathological conditions, coupling with hormone or even therapeutically relevant drug concentrations could then be compared fairly. The data are already collected by the authors and assembling a table illustrating this comparison should not be too difficult. This sort of interpretation/presentation should be quite informative.

We thank the reviewer for this comment. The information requested by the reviewer can be found in the Supplementary File 2C. As can be seen in the table, although in many cases the potency for the novel couplings is lower, this is not a universal finding since for some receptors, the pEC_50_s for the new couplings (non-canonical couplings) are similar to those of the canonical ones (for example: G_12_ for CB_1_; G_13_ for 5-HT_2C_; G_12/13_ for A_2A_ and EP_1_; G_i/o_ for CRFR1, ET_A_ and GPR39). Interestingly, in a few cases, it is the potency for the non-canonical pathways that is higher (for example: G_z_ for 5-HT_2B_; G_15_ for A_3_ and MC3R; G_12_ for B_2_, CCK_1_, CCR6 and ET_A_; G_12/13_ for CRFR1 and GPR68). A paragraph has now been added in the manuscript (lines 229-246) to discuss the implications of these potency differences for the potential physiological roles of the newly identified couplings. A note concerning the differences between the members of the same family was also made (lines 267-269). Also, the results have been discussed considering that for most receptors, the potency toward βarrestins is lower than for their canonical G protein, an observation that has also been made in the past using different assays and that is consistent with the role of βarrestins into signaling arrest at the plasma membrane.

9. The normalization procedure that compares all responses as a percentage of baseline may have unintended consequences for receptors with high constitutive activity or where endogenous ligands may be present in HEK cell cultures (e.g. adenosine, 5-HT, glutamate). Basal responses may be high for the most efficient G proteins but lower for secondary couplers, which could paradoxically give the latter a larger response window. This might skew the results, e.g. does A1 really activate G12 better than Gi subunits, or rather does the G12 sensor simply have a higher window due to lower baseline. This might also explain why Gi looks like a better coupler to mGluR5 than Gq; perhaps the Gq signal is already close to saturation, as glutamate is also difficult to remove.

The point raised by the reviewer is valid and important and we had already acknowledged this possibility in the original manuscript (lines 509-515 of the revised manuscript):

‘Finally, because EMTA is able to detect constitutive activity, high receptor expression level may lead to an elevated basal signal level that may obscure an agonist-promoted response. Such an example can be appreciated for the A_1_ receptor for which the agonist-promoted Gα_i2_ response did not reach the activation threshold criteria because of a very high constitutive activity level (Figure 5A). The impact of receptor expression on the constitutive activity and the narrowing on the agonist-promoted response is illustrated for Gαq activation by the 5-HT_2C_ (Figure 5—figure supplement 1B).’

It should however be noted that although such increased in the basal activity (whether originating from the intrinsic constitutive activity of the receptors or from the endogenous presence of agonists in the media) could affect the maximal agonist-mediated response, it should not dramatically affect the potency determined. Yet, the reduced agonist-promoted response could introduce a bias in the evaluation of the relative efficacies for different pathways (a phenomenon which is true for any assay). Because we did not asses the extent of constitutive activity of each receptor for each pathway, and given that inverse agonists, which would be needed to unambiguously conclude to constitutive activity, do not exist for all receptors, we believed that stating this possible limitation in the manuscript (lines 509-515) is sufficient to alert readers to this potential caveat.

In addition to the examples of A_1_ and 5-HT_2C_ already presented in the original submission, prompted by the reviewer’s comment, we assessed whether a higher basal activity for the G_q_ pathway *vs.* G_i2_ for mGluR5 could explain the difference in the agonist-promoted response window. As shown in Author response image 1, the basal activities detected were very similar for G_q_ and G_i2_ and the greater window observed for G_i2_ was also observed when examining the non-normalized data. Not to detract for the main message and not to hamper the flow of the manuscript, we elected not to add this data in the main manuscript.

**Author response image 1. sa2fig1:** 

10. Line 418 the authors claim that EMTA does not require amplification. It's unclear precisely what they mean in this context, but amplification could certainly occur with this assay system, as a single active receptor could maintain many G proteins in the active state. This might be especially true if the EMTA probes themselves prevent binding of RGS proteins, thus prolonging the active lifetime of each G protein. It is now clear that assays that lack amplification are advantageous for studies of ligand bias, therefore the authors should be careful to avoid making this claim.

The reviewer is right that, since one receptor can activate multiple G proteins, some level of amplification similar to those observed for assays monitoring the dissociation of the Gα from Gβγ subunits would occur with EMTA. Given that receptors can activate many G proteins before being inactivated, there are very few assays monitoring GPCR signaling that have no amplification factor. The comment made at line 418 of the original manuscript was in respect to assays that relies on the enzymatic activity of a downstream effectors such as adenylyl cyclase or phospholipase C or artificial detection systems such as gene-reporter assays or TGF-α shedding assay, which would lead to higher level of amplification than EMTA. To avoid confusion, we clarified our thinking by modifying the sentence lines 426-432.

11. Line 505 the authors refer to the use of EMTA for HTS. BRET is rarely used for true HTS, so this statement might be softened somewhat.

In fact, BRET is now regularly used in HTS formats both in academic HTS centers and in biopharmaceutical companies. In the HTS platform of IRIC alone, BRET-based assays have been used in more than 25 screens. Domain Therapeutics (DT) has also run around 50 screens and hit-to-lead campaigns in miniaturized assay format. Further, DT have profiled over 200 variants per receptor, for 60 receptors, each using 4 to 8 biosensors in full concentration-response curves, using the EMTA platform. Some examples of such screens have already been published (Hugo Lavoie et al., Nat Chem Biol. 2013; Leguay et al., J Cell Sci. 2021; Giubilaro et al., Nat Commun. 2021). It should also be noted that the Z’ obtained for many BRET-based assays, including EMTA, are larger than 0.6 and thus sufficiently robust for HTS. An example of Z’ obtained for the histamine 1 (H_1_) receptor using the combined G_z_/G_15_ sensor performed in 384-wells plate is provided for the reviewer’s perusal (Author response image 2). We respectfully think that this is sufficient to mention that the assays would be amenable for HTS. Yet, to be more conservative, we replaced ‘…may be used for high throughput screening’ by ‘…could be amenable for high throughput screening’ (line 531).

12. The authors should directly address a recent study showing that PDZ-RhoGEF is activated by Gi subunits (https://www.biorxiv.org/content/10.1101/2021.07.15.452545v1.full). Were all experiments with their PDZ-RhoGEF sensor done in the presence of PTX, as in Figure 2C?

In the original manuscript, we reported that our PDZ-RhoGEF sensor detected G_12/13_ activation but not G_i/o_ activation. Indeed, when using ET_A_ receptor, which was found to activate all G protein families, we only detected the PDZ-RhoGEF sensor activity upon heterologous expression of Gα_12_ or Gα_13_ but none of other G proteins subtypes, including G_i/o_ (see Figure 2—figure supplement 2). The selectivity of our PDZ-RhoGEF for G_12/13_
*vs.* G_i/o_ is also illustrated by the fact that many receptors that strongly activate members of the G_i/o_ family did not promote a PDZ-RhoGEF response. The apparent paradox between the results of Chandan *et al.,* indicating that PDZ-RhoGEF can be activated by G_i/o_ family members and the selectivity of our PDZ-RhoGEF construct for G_12/13_ may result from the fact that, in contrast to the full length PDZ-RhoGEF used by Chandan *et al.*, we used a truncated version of PDZ-RhoGEF that only contains the G_12/13_ binding domain and lacks the PDZ domain involved in protein-protein interaction, the actin-binding domain and the DH/PH domains involved in GEF activity and RhoA activation (Mohamed Aittaleb et al., Molecular Pharmacology 2010). The domain of PDZ-RhoGEF involved in the G_i/o_-mediated activation proposed by Chandan *et al.,* remains undetermined and may be different than the G_12/13_ binding domain of PDZ-RhoGEF used in the present study. A small sentence referring to the preprint of Chandan *et al.,* and discussing the fact that we did not detect the recruitment of our PDZ-RhoGEF upon G_i/o_ activation has been added on lines 152-158. Given the selectivity of our PDZ-RhoGEF sensor for G_12/13_
*vs.* G_i/o_ and our finding that CB_1_-mediacted activation of G_12_/G_13_ was independent of G_i/o_ protein family, since the response was not affected by PTX (Figure 2C), PTX was omitted from further G_12_/G_13_ assessments.

13. The authors claim that EMTA allows for imaging of the spatiotemporal activation of the different Biosensors. Although this is convincing for video 2 and video 3, it is less so for video 1 (galphaI). Can the authors provide additional information or an alternative representative video?

An alternative representative video for the G_q_-activation biosensor (video 1) using the AT_1_ receptor has been generated, providing a more convincing example than the original one.

14. On lines 300-301, the authors mention that HEK293 cells do not express Galpha15 and that the EP2 and opioid receptors do not couple to Gq/11 family members but couple to Galpha15. However, in Figure 3 supp 3B, there is a clear calcium response upon PGE2 or Dynorphin exposure. Galpha15 expression only potentiated the response. How is calcium signaling activated if it is not Galphaq/11 mediated?

Calcium signal has been shown to result from many distinct signaling cascades. For instance, activation of PLC by Gβγ originating from G_i/o_ activation has been well documented (WJ Koch et al., JBC 1994; MF Ethier and JM Madison, Am. J. Respir. Cell Mol. Biol., 2006; ZG Gao. and KA Jacobson, Biochem. Pharmacol. 2016; EM Pfeil, et al., Molecular Cell 2020). This could explain the calcium response observed upon κOR activation by Dynorphin A in the absence of overexpressed G_15_ in Figure 3—figure supplement 4 (of the revised manuscript). G_s_-mediated calcium entry has also been described (Zhang et al., J Pharmacol Exp Ther 2001; Christ et al., Br J Pharmacol 2009; Benitah et al., J Mol Cell Cardiol 2010) and could explain the EP_2_ response in the absence of overexpressed G_15_. It should be emphasized that the experiment described in Figure 3—figure supplement 4B was to confirm the G_15_ coupling observed with EMTA using the calcium response promoted by heterologous expression of G_15_ as an orthogonal assay and not to assess other pathways that could be involved in the calcium response evoked by κOR and EP_2_.

15. The authors suggest that the cross-activation of the D2 and alpha2 receptors by noradrenaline and serotonin is direct (through direct engagement of D2 and alpha2) as the response is inhibited by D2 and alpha2 antagonist. This conclusion would be further strengthened if the response was not inhibited by adrenergic or serotoninergic antagonists (as done with the muscarininc antagonist for CB1-CB2 cross talk).

As requested by the reviewer, we took the example of dopamine and α_2A_AR to illustrate the direct activation of a receptor by a ligand other than its natural ligand. Direct activation of the α_2A_AR by dopamine was confirmed by showing that treatment with the D_2_-family receptor selective antagonist eticlopride had negligible effect on dopamine-mediated activation of Gα_i2_ and Gα_oB_ in cells heterologously expressing α_2A_AR. In contrast, eticlopride blocked the activation of Gα_i2_ and Gα_oB_ in cells heterologously expressing D_2_. These new results are now shown in Figure 6—figure supplement 2 and discussed in the Results section of the revised manuscript (lines 384-390). Because serotoninergic antagonists are notorious for the activity on different catecholamine receptors (Roth *et al.*, Nat Rev Drug Discov. 2004), conducting a similar experiment for the serotonin-mediated activation of D_2_ and α_2A_AR would have been hazardous. We therefore, toned-down the conclusion regarding this example.

Reviewer #1 (Recommendations for the authors):There are some issues that should to addressed by the authors. None are deal breakers but they should be addressed, most likely possible without any further experimentation.The authors nicely state the caveats with the EMTA approach stating that the assays require overexpression of both G protein isoforms and receptors, although can be used with endogenous receptors. As well, they state that the assays are limited to a subset of effectors. The authors still need to address whether overexpression of the effector-sensor also biases the response. Typical endogenous effectors are expressed at very low levels owing to their roles as catalysts. In this assay, however, overexpression of the sensor is required as a BRET pair. Thus, are the authors confident that over expression of the effector-sensor does not bias the coupling response, either by altering the potency or efficacy of agonist activation?

See response to point 3 of the Essential

The coupling results are interesting with the adenosine receptors. It appears that most of the adenosine receptors tested displayed high constitutive activity. Is this due to adenosine released in the media? Studying adenosine receptors in HEK cells is not straight forward without testing the effects of adenosine deaminase to metabolize adenine-containing molecules released into the media by the cells themselves. The authors did consider the effects of glutamate in media for the mGluR studies but I'm not certain they considered it for the adenosine receptors.

See response to point 6 of the Essential

Also, when discussing the coupling efficiency of GPCRs to subsets of Galpha subunits, the authors highlight the adenosine A2A receptor as one which preferentially couples to Galpha15. The data in the table really suggest that the A2B receptor might be more selective for Galpha15 then A2A.

See response to point 7 of the Essential

To compare the receptor coupling efficiencies the authors normalized the dose-response curves of the agonists for each G protein (or arrestin) against a reference receptor on that given pathway. Similarly, the authors illustrated the Emax for each coupling relationship as a normalized response to a reference receptor that yields a maximal response. These are rational interpretations for comparison purposes. For physiological and pharmacological purposes it might be useful if the authors expressed the coupling specificity in a slightly different manner. It would be useful for readers to see how an effector response would be for a non-canonical coupling event relative to a cognate coupling relationship. For example, what would be the effector response for norepinephrine stimulation of beta1AR with the non-cognate G proteins using an EC50 concentration determined from the Galphas (cognate) measurements ? Here the physiological concentrations of norepinephrine that are necessary for Galphas activation (coupling) can be compared to the coupling to other Galphas (or arrestin) at the same concentration. At least under non-pathological conditions, coupling with hormone or even therapeutically relevant drug concentrations could then be compared fairly. The data are already collected by the authors and assembling a table illustrating this comparison should not be too difficult. This sort of interpretation/presentation should be quite informative.

See response to point 8 of the Essential

Reviewer #2 (Recommendations for the authors):The authors are for the most part clear about the fact that Gs needed to be modified in their experiments, but in several places they gloss over this in a way that tends to overstate their results. For example, on line 219 they refer to theirs as the first dataset to use "unmodified GPCRs and G proteins" in the same sentence as promiscuous coupling to 2,3 or 4 families. Some of this promiscuous coupling must have been demonstrated with modified Gs.

See response to point 5 of the Essential.

The authors refer to a companion paper that presents a detailed analysis of selectivity profiles and comparison with existing data sets. It would be better if this analysis (or at least the main points) was simply included in the present manuscript, as this might better demonstrate cases where the EMTA assays outperform other assays, a key claim of the present manuscript.

See response to point 1 of the Essential.

The authors say that their "comparative analysis" (line 275) revealed a number of couplings not found in IUPHAR or Inoue, but simple inspection of the data does the same thing. For completeness, that should at least mention that their dataset also lacks some couplings found in other datasets.

See response to point 1 of the Essential.

The term "therapeutically-relevant" is not defined, isn't necessary, and seems a stretch for many of the receptors sampled here, many of which are not demonstrated therapeutic targets.

Looking back at the list, we feel confident that all receptors studied are either already the target of clinically used drugs (74 receptors reported in Drugbank database), presently considered for the pre- or clinical development of drugs (6 receptors are currently investigational in clinical trials according to ClinicalTrials.gov database) or directly involved in specific pathophysiological processes. For more clarity, the term "therapeutically-relevant" is now defined lines 192-195 of the revised manuscript and receptors with approved drug or currently investigational in clinical trials reported in Drugbank database are now identified in the Supplementary File 2A.

The authors make frequent use of the term "engage". This is semantics, but the term is not as precise as "bind" or "activate", and seems closer to "bind". In this case I much prefer "activate", after all a key advantage of the EMTA format is that G proteins must be activated, not just bound or engaged by a GPCR.

We thank and agree with the reviewer and the term ‘engage’ has been changed for ‘activate’ in all cases except when describing the non-productive engagement of G_12_ by the V_2_ that we have changed for ‘non-productive binding’ (line 462 of the revised manuscript).

The authors use A1 adenosine receptors to demonstrate constitutive activity, with larger inverse agonist responses than agonist responses. However, a limitation of this example is the ubiquity of adenosine in cell cultures, which often requires adenosine deaminase treatment to remove completely. A better example might be another receptor with known constitutive activity (e.g. beta2AR or CB1), where the possibility of an endogenous ligand being present is more remote.

See response to point 6 of the Essential

Line 418 the authors claim that EMTA does not require amplification. It's unclear precisely what they mean in this context, but amplification could certainly occur with this assay system, as a single active receptor could maintain many G proteins in the active state. This might be especially true if the EMTA probes themselves prevent binding of RGS proteins, thus prolonging the active lifetime of each G protein. It is now clear that assays that lack amplification are advantageous for studies of ligand bias, therefore the authors should be careful to avoid making this claim.

See response to point 10 of the Essential

Line 505 the authors refer to the use of EMTA for HTS. BRET is rarely used for true HTS, so this statement might be softened somewhat.

See response to point 11 of the Essential

The authors should directly address a recent study showing that PDZ-RhoGEF is activated by Gi subunits (https://www.biorxiv.org/content/10.1101/2021.07.15.452545v1.full). Were all experiments with their PDZ-RhoGEF sensor done in the presence of PTX, as in Figure 2C?

See response to point 12 of the Essential

The normalization procedure which compares all responses as a percentage of baseline may have unintended consequences for receptors with high constitutive activity or where endogenous ligands may be present in HEK cell cultures (e.g. adenosine, 5-HT, glutamate). Basal responses may be high for the most efficient G proteins but lower for secondary couplers, which could paradoxically give the latter a larger response window. This might skew the results, e.g. does A1 really activate G12 better than Gi subunits, or rather does the G12 sensor simply have a higher window due to lower baseline. This might also explain why Gi looks like a better coupler to mGluR5 than Gq; perhaps the Gq signal is already close to saturation, as glutamate is also difficult to remove.

See response to point 9 of the Essential

Reviewer #3 (Recommendations for the authors):1) The authors highlight that one of the strengths of EMTA is the use of untagged receptor and G proteins but fail to present an example where this constitutes a true benefit when compared to other biosensors or signaling assays.

See response to point 1 of the Essential

2) Although EMTA functions in some endogenous cases where receptor expression and effector coupling are optimal, in most contexts overexpression of both GPCRs and G proteins is required. It is unclear how overexpression of GPCRs and G proteins could influence the engagement of downstream effectors and possible mislead interpretation about potential biased signaling or polypharmacology. The authors need to present a careful analysis of the relationship between receptor and G proteins overexpression (using qPCR) and effector engagement? This could be done by varying receptor and g protein expression and monitoring effector engagement.

See response to point 2 of the Essential

3) The authors have previously claimed that ligand biased signaling is dependent on cellular contexts, where various levels of expression of receptor, G proteins, effectors are found. With this in mind, how valuable is the information presented since the measurements were acquired in a non-natural context?

See response to point 2 of the Essential

4) In figure 2B on the right, it appears that PTX leads to a significant inhibition of the G protein coupling (non Galphai) in response to GnRH, is this significant? The authors state on lines 128-129 that the response is not affected by PTX. This should be clarified.

Although there is a tendency for reduction of G protein activation in presence of PTX, no statistically significant difference was found compared to untreated cells (p=0.077, 0.0636 and 0.073 using Unpaired t-test for G_q_, G_11_ and G_14_, respectively). The term ‘not significantly’ and p values have been added to the sentence (lines 124-125 of revised manuscript).

5) The authors state that effector coupling between G proteins from the same subclass are observed (i.e. different Galphai, G12 vs G13 etc…)? In order to claim this, the authors need to carefully monitor the level of expression of the G proteins (if not possible by WB then qPCR). For example, in figure 2C, the authors present data that CB1 seems to couple to Galpha 13 more than Galpha 12, two G proteins that have been thought to be functionally equivalent. Could G protein isoforms have different affinities for the biosensors?

See response to point 4 of the Essential

6) The authors claim that EMTA allows for imaging of the spatiotemporal activation of the different Biosensors. Although this is convincing for video 2 and video 3, it is less so for video 1 (galphaI). Can the authors provide additional information or alternative representative video?

See response to point 13 of the Essential

7) On lines 300-301, the authors mention that HEK293 cells do not express Galpha15 and that the EP2 and opioid receptors do not couple to Gq/11 family members but couple to Galpha15. However, in Figure 3 supp 3B, there is a clear calcium response upon PGE2 or Dynorphin exposure. Galpha15 expression only potentiated the response. How is calcium signaling activated if it is not Galphaq/11 mediated?

See response to point 14 of the Essential

8) The authors suggest that the cross-activation of the D2 and alpha2 receptors by noradrenaline and serotonin is direct (through direct engagement of D2 and alpha2) as the response is inhibited by D2 and alpha2 antagonist. This conclusion would be further strengthened if the response was not inhibited by adrenergic or serotoninergic antagonists (as done with the muscarininc antagonist for CB1-CB2 cross talk).

See response to point 15 of the Essential

9) Perhaps the most interesting new biology uncovered in this study is the indirect activation of CB1 and CB2 signaling by muscarinic agonist, a mechanism the authors suggest could involve the secretion of a cannabinoid ligand by muscarinic receptor activation. The authors however stop short of directly demonstrating that this is the case.

We agree with the reviewer that the identification of the indirect activation of CB_1_ and CB_2_ signaling by muscarinic agonist is particularly interesting and deserve a deeper investigation of the mechanisms involved. However, we believe that this goes beyond the scope of the present manuscript which is to present a new platform with the description of the signal profiling of 100 GPCRs and informs the community on how the platform could be used to detect different pharmacological phenomenon including crosstalk between receptors.